



**Global O₃-CO Correlations in a Chemistry and Transport**
**Model during July–August: Evaluation with TES Satellite**
**Observations and Sensitivity to Input Meteorological Data**
**and Emissions**
Hyun-Deok Choi[1], Hongyu Liu[1], James H. Crawford[2], David B. Considine[2,3], Dale J.
Allen[4], Bryan N. Duncan[5], Larry W. Horowitz[6], Jose M. Rodriguez[5], Susan E. Strahan[5,7],
Lin Zhang[8,9], Xiong Liu[8], Megan R. Damon[5,10], and Stephen D. Steenrod[5,7]
[1]National Institute of Aerospace, Hampton, VA
[2]NASA Langley Research Center, Hampton, VA
[3]Now at NASA Headquarters, Washington, D.C.
[4]University of Maryland, College Park, MD
[5]NASA Goddard Space Flight Center, Greenbelt, MD
[6]NOAA Geophysical Fluid Dynamics Laboratory, Princeton, NJ
[7]Universities Space Research Association, Columbia, MD
[8]Harvard University, Cambridge, MA
[9]Now at Peking University, Beijing, China
[10]Science Systems and Applications, Inc., Lanham, MD
**Submitted to** *Atmos. Chem. Phys.*, **2016**
**Correspondence:**
Hongyu Liu
Chemistry and Dynamics Branch, Mail Stop 401B
NASA Langley Research Center, Hampton, VA 23681
Tel: 757-864-3191; Email: Hongyu.Liu-1@nasa.gov



**Abstract.** We examine the capability of the Global Modeling Initiative (GMI) chemistry and
transport model to reproduce global mid-tropospheric (618hPa) $O_3$-CO correlations determined by
the measurements from Tropospheric Emission Spectrometer (TES) aboard NASA's Aura satellite
during boreal summer (July–August). The model is driven by three meteorological data sets
(fvGCM with sea surface temperature for 1995, GEOS4-DAS for 2005, and MERRA for 2005),
allowing us to examine the sensitivity of model $O_3$-CO correlations to input meteorological data.
Model simulations of radionuclide tracers ($^{222}$Rn, $^{210}$Pb, and $^{7}$Be) are used to illustrate the
differences in transport-related processes among the meteorological data sets. Simulated $O_3$ values
are evaluated with climatological ozone profiles from ozonesonde measurements and satellite
tropospheric $O_3$ columns. Despite the fact that the three simulations show significantly different
global and regional distributions of $O_3$ and CO concentrations, all simulations show similar
patterns of $O_3$-CO correlations on a global scale. These patterns are consistent with those derived
from TES observations, except in the tropical easterly biomass burning outflow regions.
Discrepancies in regional $O_3$-CO correlation patterns in the three simulations may be attributed to
differences in convective transport, stratospheric influence, and subsidence, among other
processes. To understand how various emissions drive global $O_3$-CO correlation patterns, we
examine the sensitivity of GMI/MERRA model-calculated $O_3$ and CO concentrations and their
correlations to emission types (fossil fuel, biomass burning, biogenic, and lightning $NO_x$
emissions). Fossil fuel and biomass burning emissions are mainly responsible for the strong
positive $O_3$-CO correlations over continental outflow regions in both hemispheres. Biogenic
emissions have a relatively smaller impact on $O_3$-CO correlations than other emissions, but are
largely responsible for the negative correlations over the tropical eastern Pacific, reflecting the fact
that $O_3$ is consumed and CO generated during the atmospheric oxidation process of isoprene under





low $NO_x$ conditions. We find that lightning $NO_x$ emissions degrade both positive correlations at
mid-/high- latitudes and negative correlations in the tropics because ozone production downwind
of lightning $NO_x$ emissions is not directly related to the emission and transport of CO.  Our study
concludes that $O_3$-CO correlations may be used effectively to constrain the sources of regional
tropospheric $O_3$ in global 3-D models, especially for those regions where convective transport of
pollution plays an important role.

## 1. Introduction

9        Ozone ($O_3$) is an important greenhouse gas in the troposphere and a pollutant at the surface.

It is a primary source of the hydroxyl radical (OH), which controls the oxidizing power of the
troposphere. Ozone in the troposphere is produced by photochemical oxidation of carbon
monoxide (CO), methane, and volatile organic hydrocarbons (VOCs) in the presence of nitrogen
oxides ($NO_x \equiv NO + NO_2$). Its precursors are emitted by human activity (e.g., fossil fuel
combustion and industrial processes), biomass burning, vegetation, soils, and lightning. Ozone is
also transported down from the stratosphere by the Brewer Dobson circulation. Carbon monoxide
is a product of incomplete combustion. Its sources include fossil fuel and biofuel combustion,
biomass burning, and chemical production from atmospheric oxidation of methane, isoprene, and
other VOCs. Its primary sink is reaction with OH.

19        Since CO is a precursor of tropospheric $O_3$ and an excellent tracer for long-range transport

of pollution owing to its tropospheric lifetime of a few months, correlations between $O_3$ and CO
are useful indicators of the efficiency of $O_3$ production and export (e.g., Parrish et al., 1993; Mao
and Talbot, 2004). Generally, a positive correlation in summer indicates strong photochemical





production of $O_3$ downwind of polluted regions (Chin et al., 1994; Tsutsumi and Matsueda, 2000).
A negative correlation indicates stratospheric influence (Parrish et al., 1998; Hsu et al., 2004),
photochemical $O_3$ destruction (Fishman and Seiler, 1983; Parrish et al., 1998; Mao and Talbot,
2004), or chemical production of CO (Chin et al., 1994; Real et al., 2008). Small correlation
coefficients and small linear regression slopes are indications of fresh pollution plumes that have
not yet realized their $O_3$ production potential due to, for example, incomplete photochemical
processes (Naja et al., 2003).
Many studies have used surface and/or aircraft observed $O_3$-CO correlations to understand
anthropogenic influences on $O_3$, especially in the Northern Hemisphere (NH) continental outflow
regions such as the northeastern US/northern Atlantic (Fishman and Seiler, 1983; Anderson et al.,
1993; Parrish et al., 1993; Chin et al., 1994; Fehsenfeld et al., 1996; Parrish et al., 1998; Li et al.,
2002; Honrath  et al., 2004; Mao and Talbot, 2004) and the Asian Pacific Rim (Tsutsumi and
Matsueda, 2000; Mauzerall et al., 2000). The studies found strong $O_3$-CO correlations in outflow
regions and concluded that the export of pollution from the NH major continents makes a
significant contribution to total tropospheric $O_3$ over the NH during summer.
The observed $O_3$-CO correlation coefficients and linear regression slopes have been used
to evaluate the capability of models of chemistry and transport to produce proper $O_3$ levels from
its precursors for the right reasons (e.g., Chin et al., 1994; Mauzerall et al., 2000; Zhang et al.,
2006; Voulgarakis et al., 2011). Shim et al. (2009) examined the Mexico City pollution outflow
using $O_3$, CO, and their correlations from TES as well as aircraft measurements obtained during
the Megacity Initiative: Local and Global Research Observations (MILAGRO) / Intercontinental
Chemical Transport Experiment (INTEX-B) campaigns. These investigations found that TES data
is characterized by smaller $O_3$-CO correlation coefficients but larger linear regression slopes than



*in situ* observations at 618 hPa partly due to the lack of variability in TES CO. Two previous
studies also examined TES global $O_3$-CO correlations. Zhang et al. (2006) compared mid-
tropospheric TES $O_3$-CO correlations in July 2005 over the eastern United States with those from
the GEOS-Chem model and International Consortium for Atmospheric Research on Transport and
Transformation (ICARTT) aircraft observations (July 2004), finding that TES can provide good
information on global mid-tropospheric $O_3$-CO correlations. Voulgarakis et al. (2011) expanded
the scope of Zhang et al. (2006), evaluating $O_3$-CO correlations simulated by two independent
models against five years of TES observations. They suggested that in addition to $O_3$
photochemical processes, transport may also play an important role in the $O_3$-CO correlations.
However, Voulgarakis et al. (2011) could not isolate the effect of transport because the two models
in their study used different input meteorological data sets as well as different chemical and
transport mechanisms.

13        In this paper, we present $O_3$ and CO simulations using the Global Modeling Initiative

(GMI) chemistry and transport model (CTM) driven by three meteorological data sets. The model
can incorporate different inputs and components (e.g., meteorological fields, emission inventories,
and chemical mechanisms), allowing us to test the sensitivity of model simulations to input
meteorological data sets (e.g., Douglass et al., 1999; Considine et al., 2005; Liu et al., 2016). Model
simulations are evaluated using ozonesonde and satellite observations. We then test the model's
capability to reproduce the mid-tropospheric $O_3$-CO correlations determined from TES
measurements. We present the differences in the simulated $O_3$-CO correlations due to the use of
different meteorological input data, and interpret those differences in terms of transport using
radionuclide species ($^{222}$Rn, $^{210}$Pb, and $^{7}$Be) as tracers of atmospheric transport (see Section 2.1.3).



We also investigate the effect of emission types on $O_3$ and CO concentrations and their correlations in the model.

This paper is organized as follows. Section 2: Descriptions of the GMI model, meteorological data sets, and observational data sets. Section 3: Presentation of the model simulations of radionuclides, $O_3$ and CO. Section 4: Evaluation of GMI $O_3$ and CO simulations with observations. Section 5: Evaluation of GMI $O_3$-CO correlations with satellite observations from TES. Section 6: Analysis of the effects of various emission types on the model simulated $O_3$-CO correlations. Section 7: Summary and conclusions.

## 2. Model and Data

### 2.1. GMI

#### 2.1.1. The CTM

The GMI CTM is a global 3-D composition model that combines both tropospheric and stratospheric chemical mechanisms, including 124 species, 322 chemical reactions, and 81 photolysis reactions (Ziemke et al, 2006; Duncan et al., 2007a; Considine et al., 2008; Allen et al., 2010). The tropospheric mechanism includes a detailed description of tropospheric $O_3$-NOx-hydrocarbon chemistry (Bey et al., 2001) with recent updates (e.g., see Allen et al., 2010). The stratospheric mechanism is described in Kinnison et al. (2001) and Douglass et al. (2004). The details of the GMI model are described in Duncan et al. (2007a) and Strahan et al. (2007). The basic structure of the model is described in Rotman et al. (2001).



The GMI model uses a flux-form semi-Lagrangian (FFSL) advection scheme (Lin and
Rood, 1996) and also includes parameterizations of convection, wet scavenging, dry deposition,
and planetary boundary layer mixing. The anthropogenic emission (e.g., fossil fuel emissions)
scheme is from Bey et al. (2001), Benkovitz et al. (1996), and Duncan et al. (2007b). We use the
anthropogenic emission inputs for 2005 for all simulations in this study. The biogenic emission
scheme is calculated online based on Guenther et al. (2006) and biofuel emissions are estimated
from the inventory and emission factors of Yevich and Logan (2003). Biomass burning emissions
are from Duncan et al. (2003) climatology, where the spatial and seasonal variability are derived
from satellite observations of monthly total fire counts. Lightning $NO_x$ emissions are calculated
locally in deep convection events with the scheme of Allen et al. (2010) where flash rates are
assumed to be proportional to the square of upward convective mass flux but constrained by
monthly average climatological flash rates from V2.2 of the Optical Transient Detector and the
Lightning Imaging Sensor (OTD/LIS) climatology. GMI uses modules developed at Harvard
University to calculate wet scavenging (Mari et al., 2000; Liu et al., 2001) and dry deposition rates
(Jacob and Wofsy, 1990).
Several studies have previously evaluated the GMI CTM simulations of tropospheric $O_3$
and CO. Ziemke et al. (2006) compared the tropospheric ozone columns (TOC) in an earlier
version of the GMI CTM, which was driven by the fvGCM meteorological fields (details in Section
2.1.2), with those determined from Ozone Monitoring Instrument/Microwave Limb Sounder
(OMI/MLS) measurements from the NASA Aura satellite. The comparison showed similarities
with respect to zonal and seasonal variations of TOC, but the model overestimated TOC over
northern Africa by as much as 10 DU, likely due to desert dust effects while underestimating TOC
over the western Pacific warm pool by up to 10 DU. Chandra et al. (2009) evaluated GMI TOC



when driven by the GEOS4 meteorological fields (see Section 2.1.2) with OMI/MLS TOC and found the model overestimated TOC by 5-10 DU for the latitude band 30°N - 35°N all the year and over east China in winter and spring when stratosphere-troposphere exchange (STE) is greatest. Duncan et al. (2008) showed that the annual average surface $O_3$ concentrations in the GMI/GEOS4 simulation had a high bias of about 11%, with higher biases in summer when photochemical production is the dominant source of $O_3$. Considine et al. (2008) examined the ability of GMI/fvGCM ($4° \times 5°$) to represent the observed near-tropopause $O_3$ distributions and found that annual mean $O_3$ concentrations were biased high by 45% at the model thermal tropopause likely due to insufficient vertical resolution near the tropopause (~ 1.1 km) and/or too high vertical diffusivity.

For CO, Duncan et al. (2007a) compared GMI/fvGCM simulated tropospheric CO concentrations with NOAA Global Monitoring Division (GMD) surface observations. They showed that the model was biased low at most sites in local winter/spring likely due to overestimation of OH in the simulation when the CO burden is typically at an annual maximum. Schoeberl et al. (2006) showed that GMI/fvGCM was able to reproduce the upper troposphere/lower stratosphere (UT/LS) CO tape recorder caused by seasonal changes in biomass burning, as identified with the MLS data.

**2.1.2. fvGCM, GEOS4, and MERRA Meteorological Data Sets**

We drive the GMI CTM with three meteorological datasets from: the free-running NASA Global Modeling and Assimilation Office (GMAO) finite-volume General Circulation Model (fvGCM) for 1995, the Goddard Earth Observing System Data Assimilation System Version 4 (GEOS4-DAS) for 2005, and the Modern-Era Retrospective Analysis for Research and



Applications (MERRA) for 2005. Note that the fvGCM is the general circulation model in the
assimilation system used to generate GEOS4-DAS (Bloom et al., 2005). The native vertical
coordinate of fvGCM and GEOS4-DAS models is a generalized hybrid sigma-pressure coordinate
system with 55 vertical layers and a smooth transition between sigma in the troposphere (pressure
> 200 hPa) and pure pressure in the stratosphere (top pressure 0.01 hPa). MERRA is a NASA
atmospheric reanalysis data set from a new version of GEOS-DAS Version 5 (GEOS-5.2.0).
GEOS-5 is a system of models integrated using the Earth System Modeling Framework (ESMF).
Compared to GEOS-4, GEOS-5 adopts an analysis system developed jointly with the National
Centers for Environmental Prediction (NCEP) and a different set of physics packages for the
atmospheric GCM. MERRA has 72 vertical levels with a lid at 0.01 hPa (sigma-pressure
coordinate interface at ~177 hPa). The native horizontal resolution of all meteorological data sets
is $1° \times 1.25°$. To improve computational efficiency, we drive GMI CTM with the meteorological
data sets at a degraded resolution (2° latitude by 2.5° longitude).

14        The different convective parameterizations used to generate the meteorological data sets

alters the characteristics of convective transport of chemical species. Both fvGCM and GEOS4
use the deep convection scheme of Zhang and McFarlane (1995) and the shallow convection
scheme of Hack (1994) whereas MERRA uses a modified version of the Relaxed Arakawa-
Schubert scheme for convection (Moorthi and Suarez, 1992). **Figure 1** shows the latitude-pressure
cross sections of zonal mean convective mass fluxes averaged over three meteorological data fields
and the differences from the average during July – August. fvGCM shows the strongest shallow
convection in the Southern Hemisphere (SH) mid- and low-latitudes among the models. GEOS4
shows the strongest convection in the tropical middle troposphere. MERRA is characterized by
the weakest shallow convection in both hemispheres. MERRA has the strongest tropical



convection in the lower free troposphere, but its tropical convection is not as deep as in the others.
Shallow convection in fvGCM and GEOS4 extends to higher latitudes compared to MERRA.

### 2.1.3. Radionuclide Tracers

We conduct GMI model simulations of radionuclides ($^{222}$Rn, $^{210}$Pb, and $^{7}$Be) to examine
the relative effects of convection, stratospheric influence, and large-scale subsidence on the
transport of trace species and their sensitivity to input meteorological data sets. $^{222}$Rn has a half-
life of 3.8 days and is emitted primarily from continental crust. It is useful as a tracer of convective
transport in global models (e.g., Jacob et al., 1997). $^{210}$Pb, a decay daughter of $^{222}$Rn, has a
radioactive half-life of 22.3 years, and $^{7}$Be, which is produced by cosmic ray spallation reactions
in the stratosphere and UT, has a radioactive half-life of 53.3 days. Because $^{210}$Pb and $^{7}$Be attach
to submicron aerosols after production and are therefore scavenged by precipitation or deposited
to the surface, they have been used as a pair to test wet deposition schemes in global models (e.g.,
Liu et al., 2001). $^{7}$Be is also used as a tracer for STE (Dibb et al., 1994; Liu et al., 2001; Liu et al.,
2016). The ratio $^{7}$Be/$^{210}$Pb is useful as an indicator of vertical transport because the ratio is
insensitive to precipitation scavenging (Koch et al., 1996).

### 2.2. Data Sets

### 2.2.1. Ozonesonde O$_3$

We use climatological ozone profiles from 23 ozonesonde stations averaged over July –
August from 1985 to 2000, originally constructed by Considine et al. (2008) based on Logan (1999)
and Thompson et al. (2003). The number of soundings at each station is adequate for defining
monthly means used to evaluate the accuracy of the model results (Considine et al., 2008; Liu et
al., 2016).





**2.2.2. Satellite Tropospheric Ozone Column (TOC)**

Three TOC products are used in this study: Total Ozone Mapping Spectrometer (TOMS) – Solar Backscatter Ultraviolet (SBUV), OMI-MLS, and directly retrieved TOC from TES. The TOMS-SBUV TOC and OMI-MLS TOC are determined using the tropospheric ozone residual (TOR) method, which involves subtracting measurements of SBUV and MLS stratospheric column ozone (SCO) from TOMS and OMI total column ozone, respectively (Fishman et al., 2003; Ziemke et al., 2006). The TES TOCs are integrated from directly retrieved volume mixing ratios. We did not consider different instrument sensitivities because integrating retrievals significantly reduces the error due to averaging over pressure ranges larger than TES vertical resolution (Osterman et al., 2008; Zhang et al., 2012). Tropopause pressure is taken from the GEOS4 meteorological data (2° x 2.5°). A description of TES retrievals is given in Section 2.2.3.

**2.2.3. TES $O_3$ and CO.**

The TES instrument on EOS-Aura routinely provides observations of tropospheric $O_3$ and CO across the globe (Beer et al., 2001; Beer, 2006). The Aura satellite is on a polar sun-synchronous orbit with equator crossing at 01:45 (descending) and 13:45 (ascending) local time. TES is a Fourier transform infrared emission spectrometer with high spectral resolution (0.1 cm$^{-1}$) and wide spectral range (650 – 3050 cm$^{-1}$) (Beer et al., 2001). The nadir footprint of TES is $5 \times 8$ km. TES observations consist of two modes: global survey and special observations (Beer et al., 2001). We use TES level 2, version 4 global survey nadir observations (http://eosweb.larc.nasa.gov/) and only $O_3$ and CO retrievals with the "Master" quality flag are used in this analysis. The retrievals of $O_3$ have 1 – 1.5 degrees of freedom (DOF) in the profile at mid-latitudes in summer, with peak sensitivities near 700 hPa and 300-400 hPa, respectively



(Parrington et al., 2008). TES CO profiles generally have $1 - 1.5$ DOFs in the troposphere (Luo et
al., 2007ab). Detailed descriptions of the TES instrument and the $O_3$ and CO retrieval algorithms
are described in Beer et al. (2001, 2006), Worden et al. (2004), and Bowman et al. (2002, 2006).

4        In this study, we use $O_3$ and CO retrievals at 618 hPa level, where TES has good sensitivity

for both $O_3$ and CO centered in the MT, and exclude latitudes $> 60°$ where TES measurements are
less reliable due to low brightness temperatures (Zhang et al., 2006). Due to limitation of TES
vertical resolution ($1 - 1.5$ DOFs in the troposphere for both $O_3$ and CO), TES averaging kernels
are applied to the simulations to take into account the different sensitivities of the instruments.
TES uses MOZART model output binned by month and in blocks of $10°$ latitude by $60°$ longitude
as a priori profiles (Worden et al., 2004). Validation of TES $O_3$ against ozonesondes showed that
TES ozone typically has a high bias of about 10% in the UT (Worden et al., 2007) or 3-10 ppbv in
the MT/UT (Nassar et al., 2008). TES CO has a negative bias (<10%) compared to aircraft
measurements in the NH mid-latitude LT/MT during the INTEX-B mission (spring 2006) (Luo et
al., 2007a).
**3. Model Simulations of Radionuclides, $O_3$, and CO**
**3.1. GMI Simulations of Radionuclides**

18       **Figures 2 and 3** show the latitude-pressure cross sections of zonal mean concentrations of

$^{222}$Rn and stratospheric fraction (%) of tropospheric $^7$Be concentrations during July – August for
the values averaged over three meteorological data sets and the differences from the mean.
Differences in zonal mean $^{222}$Rn concentrations at SH mid-latitudes among the three simulations
are small despite much stronger shallow convection in fvGCM (**Figure 1**). This reflects the fact



that most convection at SH mid-latitudes occurs over the ocean. However, GMI/fvGCM $^{222}$Rn
concentrations in the UT at NH subtropics and mid-latitudes are ~ 20 − 70 % higher than those in
other simulations due to the deeper convection in fvGCM (**Figure 1**). In the tropical UT/MT,
GMI/MERRA produced the lowest $^{222}$Rn concentrations, consistent with the lower cutoff of
convection in GMI/MERRA (**Figure 2**). This is not inconsistent with the largest stratospheric
influence in the tropical UT/MT in GMI/MERRA among the three meteorological data sets
(**Figure 3**). Previously, Liu et al. (2010) and Zhang et al. (2011) used GEOS-Chem simulations of
CO and $^{222}$Rn (driven by GEOS4 DAS and GEOS5 DAS meteorological data) to show that the
tropical convection in GEOS4 is deeper than in GEOS5. Because the MERRA reanalysis utilizes
the same GCM as the GEOS5 DAS, it also utilizes the Relaxed Arakawa-Schubert (RAS)
convection.

12        The stratospheric contribution to the lower-tropospheric $^{7}$Be concentrations in

GMI/fvGCM peaks near 30 - 75°N (20 − 25%), in contrast to the GMI/GEOS4 and GMI/MERRA
simulations (**Figure 3**). The GMI/GEOS4 and GMI/MERRA simulations show a similar pattern
of stratospheric influence on the troposphere with maxima near 0 − 30°S and >30°S (20 - 30%),
respectively, in the LT. However, GMI/GEOS4 suggests more stratospheric influence than
GMI/MERRA in the MT near 30°S (30 - 35%) and near 30 − 45°N (~ 25%). The stratospheric
impacts on the tropical MT/UT are weakest in GMI/fvGCM and strongest in GMI/MERRA. At
NH mid-latitudes, stratospheric influences on the LT are largest and most extensive in
GMI/fvGCM and smallest in GMI/MERRA. These differences in stratospheric influence that
characterize these meteorological data sets will be used to interpret GMI $O_3$ and CO simulations
driven by these meteorological fields (Sections 3.2 and 4).
**3.2. GMI Simulations of O₃ and CO**



1       **Figures 4-5** show the latitude-pressure cross sections of zonal mean mixing ratios of $O_3$

and CO during July – August averaged over three simulations and the differences from the mean.
The latitudinal distributions of $O_3$ from all simulations show lowest $O_3$ concentrations near the
surface at high latitudes and in the tropical LT (**Figure 4**). Relatively low $O_3$ in the tropical free
troposphere results from transport of ozone-poor air from the LT to UT/MT via deep convection.
High $O_3$ concentrations are seen in the (subtropical) descending branches of the Hadley circulation
partly due to the influence of STE. Compared with GMI GEOS4 and GMI/MERRA, GMI/fvGCM
simulates higher $O_3$ in the NH mid-latitude MT and lower $O_3$ in the SH LT/MT. This is likely due
to higher STE in the NH and weaker STE in the SH, respectively, as suggested by the higher (lower)
fraction of stratospheric $^7$Be seen in the GMI/fvGCM simulation compared to the other two
simulations (**Figure 3**). On the other hand, GMI/MERRA simulates the highest $O_3$ in the tropical
MT/UT as a result of "stronger but shallower" deep convection in the tropics (**Figure 1**). All
simulations show the largest CO concentrations in the tropical LT/MT and NH mid-latitude
boundary layer (**Figure 5**). The former reflects convective lifting of tropical biomass burning CO
emissions and the latter anthropogenic CO emissions, respectively. Among the three simulations,
GMI/fvGCM simulates the lowest CO concentrations in the tropical MT/UT as well as both
hemispheres. In the NH MT/UT in GMI/fvGCM, the low CO concentrations result from high OH
concentrations associated with high $O_3$ concentrations due to higher STE, which will be discussed
in Section 4.1. In the SH GMI/fvGCM, the low CO concentrations are due to high OH
concentrations as a result of too low $NO_x$ emissions from lightning (see Section 4.1). Tropical
MT/UT CO concentrations in GMI/MERRA are not as high as those in GMI/GEOS4, again
reflecting the "shallower" tropical deep convection in MERRA.



## 4. Evaluation of GMI O$_3$ and CO Simulations with Observations

In this section, we evaluate GMI O$_3$ and CO simulations driven by the fvGCM, GEOS4, and MERRA meteorological data sets with ozonesonde O$_3$ vertical profiles, satellite TOC, and TES O$_3$ and CO retrievals.

### 4.1. Ozone Vertical Profiles and Tropospheric Ozone Column

**Figure 6** compares GMI simulated tropospheric O$_3$ profiles with ozonesonde observations averaged over July-August for a range of latitudes. These results are typical of other stations at similar latitudes. GMI/fvGCM overestimates O$_3$ in the NH high-/mid-latitude UT/MT (e.g., Churchil, Hohenpeissenberg, and Sapporo). This may be due to excessive STE given the relatively high fractions of [7]Be from the stratosphere (**Figure 3**). The overestimate may also be partly attributed to strong convective mass fluxes in the NH mid-latitude that lift more O$_3$ and/or its precursors from the surface (**Figures 1-2**). **Figure 6** also shows that GMI/fvGCM underestimates O$_3$ in the SH (e.g., Reunion Island). Since stratospheric [7]Be fractions are relatively low in this simulation, the O$_3$ underestimate may be due to overly weak STE (cf. **Figure 3**). Low emissions of lightning NO$_x$, an important precursor of tropospheric O$_3$ could also play a role. Lightning NO$_x$ emissions between 10°S and 70°S in GMI/fvGCM during July-August are similar to those in GMI/GEOS4 and GMI/MERRA, but the emissions during May-June are a factor of ~ 2.5 lower than those in GMI/GEOS4 and GMI/MERRA (**Table 1**). Since O$_3$ has a lifetime of weeks to months in the UT/MT, a low-O$_3$ bias during May-June will lead to lower O$_3$ during July-August in GMI/fvGCM. GMI/GEOS4 simulates O$_3$ in both hemispheres reasonably well but underestimates O$_3$ in the tropical UT/MT, as seen at Paramaribo and Nairobi in **Figure 6**. GMI/MERRA underestimates O$_3$ in the NH high-latitude UT (e.g., Resolute) likely due to weak





STE compared to GMI/GEOS4 as suggested by [7]Be tracer simulations (**Figure 3**), while it
overestimates $O_3$ with a high bias in the SH subtropics (e.g., Samoa and Reunion Island) because
of a combination of excessive influences from lightning $NO_x$ emissions in May (**Table 1**) and STE
(or subsidence from UT) (**Figure 3**). In addition, the shallower tropical convection (**Figure 1**)
accompanied by larger STE contribution in the southern tropical MT/UT (**Figure 3**) results in less
clean air being lifted from the LT to MT/UT.

7         **Figure 7** shows GMI simulated zonal mean TOCs averaged over July – August in

comparison with TORs determined from TOMS/SBUV (Fishman et al., 2003), OMI/MLS
(Ziemke et al., 2006), and TOCs directly retrieved from TES measurements. The World
Meteorological Organization (WMO) definition of thermal tropopause is used to calculate the
model TOC, following Liu et al. (2016). The latitudinal distribution of TOCs shows a trough in
the tropics and polar regions, and a peak at mid-latitudes in both the models and the observations.
The TORs determined from TOMS/SBUV and OMI/MLS agree well with each other in the NH,
but those from OMI/MLS are lower at ~10ºN and higher at south of 50ºS. The TOCs determined
from TES are more similar to the OMI/MLS TORs, but biased high in the northern subtropics, and
biased low at south of 40ºS. A comparison of **Figure 7** with **Figure 6** indicates that the TOCs from
three model simulations coincide with the above results from model evaluations with ozonesonde
$O_3$ profiles. For example, both evaluations suggest negative biases in the SH and positive biases
in the NH high-/mid-latitudes in GMI/fvGCM, and positive biases in the southern subtropics in
GMI/MERRA.
**4.2. $O_3$ and CO Concentrations at 618 hPa**



**Figure 8** shows the July-August mean concentrations of $O_3$ and CO at 618 hPa in the GMI
simulations. **Figure 9** shows the corresponding global distributions of $^{222}$Rn concentrations,
stratospheric fractions (%) of mean tropospheric $^{7}$Be concentrations, and ratios $^{7}$Be/$^{210}$Pb. All
simulations show highest $O_3$ concentrations at NH mid-latitudes and lowest $O_3$ concentrations in
the tropical western Pacific. They also simulate a narrow band of relatively high $O_3$ concentrations
in the southern tropics and subtropics. GMI/fvGCM simulates highest $O_3$ concentrations at NH
mid/high latitudes (**Figure 8**, left panel) likely due to STE, as indicated by a large fraction of $^{7}$Be
transported down from the stratosphere (**Figure 9**, middle top panel). By contrast, it simulates the
lowest $O_3$ concentrations in the southern tropics and subtropics, especially over southern Africa
and South Atlantic Ocean. In this region, GMI/MERRA simulates the highest $O_3$ concentrations
attributed to high lightning $NO_x$ emissions (**Table 1**), large STE (**Figure 9**, middle bottom panel),
and biomass burning emissions lifted by shallow but strong convection (Section 3; **Figure 9**, left
bottom panel). Thompson et al. (1996) previously suggested that $O_3$ maximum observed in
southern Africa and the adjacent Atlantic during September - October 1992 is caused by the
coincidence of $O_3$ precursors from biomass burning with long residence time, and deep convection
with additional lightning $NO_x$ and biogenic sources. As we will show in Section 5, the emission
types contributing to the $O_3$ enhancements over this region in July – August mainly include
lightning $NO_x$ and, to a lesser extent, biomass burning.
All simulations show a similar pattern of CO concentrations at 618 hPa, e.g., CO
enhancements due to biomass burning emissions lifted by convection (e.g., South America, Africa,
Indonesia, and Alaska) and anthropogenic emissions (e.g., East Asia, South Asia, and eastern
North America) (**Figure 8**, right column). This pattern also reflects the geographic distribution of
these emissions. GMI/fvGCM simulates lowest CO concentrations at 618 hPa in most of the



polluted regions due to stronger STE of $O_3$, as discussed in Section 4.1. GMI/GEOS4 simulates
slightly lower CO concentrations in East and South Asia, North America and their outflow regions,
and Indonesia than GMI/MERRA does. GMI/GEOS4 also simulates lower CO concentrations over
subtropical South American and African westerly outflow regions.
To evaluate GMI $O_3$ and CO simulations with satellite observations, we use TES retrievals
at 618 hPa where TES has good sensitivity for both $O_3$ and CO in the MT (Zhang et al., 2006).
GMI model output was sampled along the TES orbit track at the observation time and then
interpolated onto the 67 vertical pressure levels of TES retrievals. Since the model output was
saved every 3 hours, the temporal offset with TES is up to 1.5 hours. To compare the model output
with the TES retrieved profiles, TES averaging kernels and *a priori* were applied to the model
output. Both the model output and TES data were gridded onto grids of 10° latitude by 10°
longitude by averaging all values within each grid box. **Figures 10-11** show the mean
concentrations of $O_3$ and CO at 618 hPa observed by TES during July - August 2005 and
corresponding GMI CTM results.
TES observed enhanced $O_3$ concentrations over the Middle East, northern Africa, southern
Africa, North America, and East Asia (**Figure 10**). Increased levels of $O_3$ were also observed in
continental outflow regions, especially the northwestern Pacific, North Atlantic, tropical south
Atlantic, and southern subtropical Indian Ocean. All simulations capture the spatial distributions
of $O_3$ well but underestimate the enhancements over southern Africa and adjacent oceans.
GMI/fvGCM simulates reasonably well the TES-observed $O_3$ enhancements at NH mid/high
latitudes but slightly underestimates the low $O_3$ concentrations in the tropical western Pacific and
Indian Ocean. GMI/GEOS4 and GMI/MERRA simulations show lower $O_3$ concentrations at NH
mid/high latitudes compared to TES observations. However, considering that TES $O_3$ has a



positive bias of 3-10 ppbv in the MT (Nassar et al., 2008), GMI/fvGCM may very well
overestimate $O_3$ at NH mid-latitudes while GMI/GEOS4 and GMI/MERRA simulations are closer
to reality. This conclusion is consistent with that from the comparison of GMI simulations with
ozonesonde observations (**Figure 6**).
Enhanced CO concentrations were observed by TES over Africa, South America, North
America, and Eurasia (**Figure 11**). All simulations underestimated CO concentrations in most of
those CO hot spots in the NH. GMI/GEOS4 captured fairly well high CO concentrations over
biomass burning regions in South America and Africa. However, considering TES CO biases, i.e.,
a negative bias at NH mid-latitudes and a positive bias in the tropics (Luo et al., 2007a; Lopez et
al., 2008), all simulations significantly underestimate CO enhancements at NH mid-latitudes but
simulate better CO enhancements over the tropical biomass burning regions. This is consistent
with a previous study by Shindell et al. (2006) who found multi-model underestimate of NH
extratropical CO likely due to current inventories underestimating fossil fuel emissions in East
Asia and biomass burning emissions in south-central Africa.

## 5. $O_3$ and CO Relationships

In this section, we examine $O_3$ and CO relationships at 618hPa in GMI CTM. We interpret
GMI simulated $O_3$-CO correlations and their slopes in the context of emissions, photochemical
transformation, and transport (e.g., convection, STE, and large-scale subsidence), using model
meteorological data and radionuclide simulations. We then evaluate them with those derived from
TES satellite observations.

### 5.1. GMI $O_3$-CO Correlations



**Figure 12** shows the $O_3$-CO correlation coefficients (R) and linear regression slopes
($dO_3/dCO$) at 618 hPa for July – August, as calculated using the reduced major axis method with
3-hourly output from the GMI/fvGCM, GMI/GEOS4, and GMI/MERRA simulations. We discuss
the common features in the correlation patterns in all simulations, followed by their discrepancies.
All simulations show strong positive $O_3$-CO correlations and large $dO_3/dCO$ enhancement ratios
in the NH major continental outflow regions, e.g., Atlantic Seaboard, northern Atlantic, and
northern Pacific, consistent with previous modeling studies (Zhang et al., 2006; Voulgarakis et al.,
2011) and in situ observations (e.g., Anderson et al. 1993; Chin et al., 1994; Jaffe et al., 1996;
Parrish et al., 1998; Tsutsumi and Matsueda, 2000; Mao and Talbot, 2004). Our simulations also
suggest a much larger area with high correlations that extends from the NW to NE Pacific. We
found that strong positive correlation regions are not co-located with maximum $O_3$ and CO
concentrations in all simulations. Instead, they are located between most polluted and clean areas,
reflecting the intrusion of high $O_3$ (and CO) air from mid-latitudes and low $O_3$ (and CO) air from
the tropics. Fishman and Seiler (1983) and Mauzerall et al. (2000) previously suggested that strong
positive $O_3$-CO correlations in low CO regions may be caused by the depletion of both $O_3$ and CO
in tropical air.
All simulations show positive $O_3$-CO correlations in the SH marine regions, but
GMI/MERRA simulates much stronger negative correlations over the equatorial Atlantic. The
latter reflects the stronger convection in the LT/MT in MERRA, which will be discussed below.
Positive $O_3$-CO correlations were previously observed over the tropical South Atlantic during the
TRACE-A aircraft mission (September–October, 1992) (e.g., Collins et al., 1996). Collins et al.
concluded that the $O_3$-CO correlations over the tropical South Atlantic are more affected by in situ



photochemical production from aged biomass burning plumes (positive $O_3$-CO correlation) than
transport from the stratosphere (negative $O_3$-CO and $O_3$ - dew point correlations).

3        Strong positive $O_3$-CO correlations are present in all simulations at 618 hPa over Indonesia

(**Figure 12**), reflecting convective transport of biomass burning CO (**Figure 8**) and photochemical
production of $O_3$ from its precursors. The $dO_3/dCO$ enhancement ratios over Indonesia are not as
large as those over the NH mid-latitude continental outflow regions due to the fact that biomass
burning emits $NO_x$ less efficiently than fossil fuel does.

8        Positive $O_3$-CO correlations over the westerly African biomass burning outflow region

(southern Indian Ocean, ~ 45ºS) are seen in all simulations (**Figure 12**). The positive $O_3$-CO
correlations over both the NH mid-latitude continental outflow regions and the westerly African
biomass burning outflow regions mainly reflect $O_3$ and CO signatures from different sources: 1)
anthropogenic emissions of CO and other $O_3$ precursors in the former and biomass burning
emissions in the latter (**Figure 8**), and 2) significant influences from the stratosphere and
subsidence from UT/LS (**Figure 9**, middle and right columns, respectively). In the case of 1), the
$dO_3/dCO$ slopes in the westerly African biomass burning outflow are smaller than those in the NH
mid-latitude continental outflow, again reflecting the lower efficiency of biomass burning $NO_x$
emissions than that of fossil fuel $NO_x$ emissions. In the case of 2), mixing of stratospheric air (high
$O_3$) with polluted air masses (high CO) has previously been found associated with positive $O_3$-CO
correlations downwind from outflow regions (Cooper et al., 2002; Kim et al., 2013).

20        Strong negative $O_3$-CO correlations are seen in all simulations over the northern tropical

eastern Pacific, Caribbean, northern tropical Atlantic, and equatorial Africa. These negative
correlations are primarily a result of convective transport of low-$O_3$ air masses impacted by





biogenic emissions. As will be discussed in Section 6, significant decreases in $O_3$ and increases in
CO occur near the above regions due to atmospheric oxidation of biogenic VOCs (e.g., isoprene)
over tropical America and Africa. In addition, our results show weak positive (in GMI/fvGCM
and GMI/GEOS4) or strong negative (in GMI/MERRA) $O_3$-CO correlations over much of the
southern tropics and subtropics, especially near the biomass burning outflow regions. Negative
$O_3$-CO correlations in the southern tropics during July-August were previously reported by
Fishman and Seiler (1983). Based on aircraft measurements, they concluded that $O_3$ destruction in
the southern tropical LT, where the major CO sources (biomass burning emissions) are located,
may lead to strong negative correlations (see their Figure 3).

10       All GMI simulations show strong negative $O_3$-CO correlations over the Asian continent

including the Middle East (**Figure 12**). Over Southwest China (e.g., Sichuan Basin), monsoonal
convective lifting of air masses with high-CO and low-$O_3$ leads to negative $O_3$-CO correlations.
For most of other regions, high $O_3$ and low CO associated with stratospherically influenced air
(**Figure 9**, middle column) result in negative $O_3$-CO correlations with large (negative) $dO_3/dCO$
ratios. As will be discussed in Section 6, lightning $NO_x$ emissions also contribute to these negative
correlations over the Asian continent. Our simulations over the Tibetan Plateau are consistent with
the study of Wang et al., (2006), who inferred negative $O_3$-CO correlations from in situ
measurements at Mount Waliguan located at the northeastern edge of the Tibetan Plateau during
summer due to downward transport from the UT/LS.

20       While the $O_3$-CO correlations in the three simulations show similarities, they also show

differences. The global $O_3$-CO correlation patterns in GMI/fvGCM and GMI/GEOS4 are more
similar, presumably because fvGCM is the GCM in the GEOS4 assimilation and they use the same
convection scheme. Even so, significantly different $O_3$-CO correlation coefficients between





GMI/fvGCM and GMI/GEOS4 are seen in northern Africa, where the former simulates strong negative but the latter shows weak positive correlations. As indicated by radionuclide tracers ($^{210}$Pb and $^{7}$Be), fvGCM has relatively stronger large-scale subsidence over northern Africa at 618 hPa than GEOS4, resulting in strong correlations with a large negative slope. In addition, the $O_3$-CO correlations in GMI/MERRA are strongly negative over northern South America, tropical western South Atlantic Ocean, Indian Ocean, and tropical western Pacific Ocean. By contrast, the correlations in these regions in GMI/fvGCM and GMI/GEOS4 are either weak or positive. The convection in fvGCM is much weaker than in GEOS4 or MERRA except at SH mid-latitudes and over Tibetan Plateau (not shown). MERRA has the strongest convection in Central America, tropical western Pacific Ocean, tropical eastern Pacific Ocean, tropical western Atlantic Ocean, tropical eastern Indian Ocean, and Bay of Bengal. These differences of convective mass fluxes result in broader regions with negative $O_3$-CO correlations in the tropics in GMI/MERRA than those in GMI/fvGCM and GMI/GEOS4. Kim et al. (2013) also simulated different $O_3$-CO correlations in some tropical regions with GEOS-Chem driven by GEOS4 and GEOS5 meteorological data sets because of the model transport error associated with deep convection.

## 5.2. Evaluation of GMI $O_3$-CO Correlations with TES Observations

**Figures 13** and **14** show the $O_3$-CO correlation coefficients (R) and linear regression slopes ($dO_3/dCO$), respectively, at 618 hPa as determined by TES observations for July - August 2005, and corresponding GMI CTM results with 3-hourly output sampled along the TES orbit tracks. Values are calculated in $10° \times 10°$ grid cells. The regions of $> 60°S$ and $> 60°N$ are excluded in this study because $O_3$ and CO concentrations over these regions are low (**Figure 8**) and absolute co-variances of $O_3$ and CO over these regions are also low (not shown). Therefore, as suggested by Voulgarakis et al. (2011), discrepancies in these regions are not scientifically important in terms





of the $O_3$-CO correlation. Since only two months of TES $O_3$ and CO observations were used, the
correlation patterns are somewhat patchy and correlations are weak ($|R| < 0.2$) over more than half
of the globe. Using TES data for July – August over 5 years improves the consistency of the
correlation patterns (**Figure 15**), as discussed later. TES-observed $O_3$ and CO concentrations show
highest correlations (R up to 0.6) with large slopes over the western Pacific, and relatively high
correlations (R = 0.2-0.4) with relatively large slopes over North America, the Middle East,
northern South America, central and southern Africa, as well as continental outflow regions, e.g.,
northwestern Pacific Ocean, western Indian Ocean, subtropical South Atlantic Ocean, tropical
eastern Pacific, and northwestern Atlantic (**Figures 13** and **14**). Negative correlations were
observed over the Tibetan Plateau (R < -0.6), northern Africa, and SH mid-latitudes (R < -0.4)
(**Figure 13**). Global TES $O_3$-CO correlation patterns and magnitudes are similar to those reported
by Zhang et al. (2006) and Voulgarakis et al. (2011). The slope patterns (**Figure 14**) follow the
correlation ones (**Figure 13**), suggesting that the slopes of the regression lines are useful indicators
of the correlation strength.
The GMI simulated $O_3$-CO correlation coefficients and linear regression slopes ($dO_3/dCO$)
calculated from each of the three model outputs sampled along the TES orbit tracks show similar
global patterns but overall weaker correlations (**Figure 13**) and smaller slopes (**Figure 14**) than
non-sampled raw model results (**Figure 12**) due to spatiotemporal sampling and application of
TES averaging kernels. All simulations capture the TES-observed positive $O_3$-CO correlations in
various regions. On the other hand, all simulations indicate strong negative correlations over the
Tibetan Plateau and tropical convective regions where TES misses such correlations or only shows
much weaker negative correlations in much narrower areas.



1        To get a more statistically robust view of TES $O_3$-CO correlations, we conduct a similar

analysis using multi-year observations. **Figure 15** shows the $O_3$-CO correlation coefficients (R)
and linear regression slopes ($dO_3/dCO$) at 618 hPa as determined by TES $O_3$ and CO retrievals for
July - August over 5 years (2005 – 2009). Values are calculated in 4°x5° grid cells. The global
distributions provide more details and are consistent with the coarse patterns for July – August
2005 shown in **Figures 13** and **14**. The negative correlations over the Tibetan Plateau and northern
Africa are more apparent than those using the TES data only for July – August 2005 (**Figure 13**).
Overall our results of multi-year (2005 – 2009) $O_3$-CO correlation coefficients at 618 hPa for July
– August are similar to those inferred from the mean mid-tropospheric (400 – 800 hPa) TES $O_3$
and CO concentrations averaged over July – August 2005 – 2008 (Voulgarakis et al., 2011).
**6. Sensitivity of $O_3$-CO Correlations to Emissions**

13       In order to understand how $O_3$-CO correlation patterns are driven by emissions, we

examine the sensitivity of $O_3$-CO correlations to emission types in the GMI model driven by the
MERRA meteorological fields, which represent the state-of-the-art of GEOS-DAS at the time of
this study. **Figures 16 - 19** show the mean changes in $O_3$ and CO concentrations (ppbv) and their
correlation coefficients, as well as the areas where correlation signs change relative to the standard
simulation at 618 hPa when each emission type (fossil fuel, biomass burning, biogenic, and
lightning $NO_x$ emissions) is excluded in the model for July – August 2005. **Figure 20** shows the
$O_3$-CO correlation coefficients (R) at 618 hPa in the standard simulation and when each emission
type is excluded. Results are calculated using 3-hourly model output. These figures provide the
context for discussions in this section.





Fossil fuel emissions substantially increase $O_3$ (by ~5-20 ppbv) and CO (by ~10-30 ppbv)
in the NH, notably over the Asian and North American continental outflow regions (**Figure 16ab**).
Fossil fuel emissions lead to strengthened $O_3$-CO correlations with correlation signs changing from
negative to positive over the Asian and North American outflow regions (**Figures 16cd** and **20b**).
Such effects are also seen over Europe, the Arabian Sea, the northern Bay of Bengal, and the
northeastern Pacific (**Figure 16cd**). Fossil fuel emissions result in stronger negative $O_3$-CO
correlations over part of the Asian continent (**Figure 16c**). This is especially true over the Tibetan
Plateau where low-level convergence transports air masses with low-$O_3$ and high-CO to the middle
troposphere.
Biomass burning emissions increase $O_3$ (by ~2-10 ppbv) and CO (by > 25ppbv) in the
easterly outflow in the tropical South America and Central Africa, in the westerly outflow in the
southern subtropics, and over Indonesia (**Figure 17ab**). They are responsible for the positive
correlations in the SH mid- and high- latitudes (**Figures 17cd** and **19c**). Without biomass burning
emissions, $O_3$-CO correlations over the westerly outflow in the southern subtropics and most of
the SH mid- and high- latitudes would be negative or very weak (**Figures 17c** and **20c**). By contrast,
biomass burning emissions degrade an already strong correlation from fossil fuel emissions in the
NH (e.g., over part of the tropical western Pacific, Bay of Bengal, NH subtropical Atlantic, and
especially NH high latitudes, **Figure 17d**). In the tropics, biomass burning emissions strengthen
the positive correlations in Indonesia and weaken the negative correlations over the tropical South
American outflow region. In the two models of Voulgarakis et al. (2011), biomass burning
emissions have the largest impact on the $O_3$-CO correlations in the tropics, especially downwind
of Central Africa and South America where biomass burning emissions changed the correlation
sign from negative to positive. Our results show no apparent changes in the $O_3$-CO correlation





signs (negative) in these downwind regions. This may reflect the differences in biomass burning
emissions and/or chemical mechanisms used in the two studies.

3        Biogenic emissions increase $O_3$ concentrations at 618 hPa by ~ 2-6 ppbv in the NH

subtropics and mid-latitudes, but decrease $O_3$ concentrations by up to ~10 ppbv in tropical South
America, tropical Africa, and Indonesia (**Figure 18a**). The latter mainly reflects the fact that $O_3$ is
consumed during the atmospheric oxidation process of isoprene under low $NO_X$ conditions (Fan
and Zhang, 2004; Seinfeld and Pandis, 1998). Biogenic emissions have large positive impacts on
CO concentrations in the easterly and westerly outflow regions of South America and Africa, in
the North American outflow, over Southwest China and Indonesia, as well as in the SH background
(**Figure 18b**). The $O_3$-CO correlations in the model show smaller sensitivity to biogenic emissions
relative to other emission types (**Figures 18c** and 20**d**). Nevertheless, biogenic emissions lead to
strong negative $O_3$-CO correlations over the tropical eastern Pacific Ocean due to reduced $O_3$ and
enhanced CO concentrations associated with these emissions (**Figures 18cd** and **20d**). Such effects
are also seen over central Africa, easterly South American outflow, westerly South American
outflow, Indonesia, and subtropical western Pacific.

16       Lightning $NO_X$ emissions increase $O_3$ concentrations at 618 hPa by up to ~15-25 ppbv at

NH subtropics and mid-latitudes, and by up to ~15-30 ppbv at SH tropics and subtropics (**Figure
19a**). Such increases are relatively larger in those regions with subsiding air from the UT (cf.,
**Figure 9**, right bottom panel), where the largest effect of lightning $NO_X$ emissions occurs. The
resulting increase in OH concentrations leads to a general decrease in CO concentrations with
maximum effects in the tropics and SH subtropics (**Figure 19ab**). Consequently, lightning $NO_X$
emissions weaken both the positive $O_3$-CO correlations at mid- and high- latitudes and the negative
correlations in the tropics (**Figures 19cd** and **20e**). They alter the correlation signs from positive



to negative in various areas where the correlations are generally weak (**Figure 19d**). Our results
are in contrast with those of Voulgarakis et al. (2011) who showed that lightning $NO_X$ emissions
appeared to increase the $O_3$-CO correlations (400-800 hPa) in various regions (e.g., tropical eastern
Pacific, NH continental outflow regions). These may partly reflect the differences in the altitude
and strength of lightning $NO_x$ emissions.
**7. Summary and Conclusions**

8         We have examined the capability of the Global Modeling Initiative (GMI) chemistry and

transport model (CTM) to reproduce the global mid-tropospheric $O_3$-CO correlations from the TES
instrument onboard the NASA Aura satellite during boreal summer (July – August). The model
was driven by three meteorological data sets (fvGCM for 1995, GEOS4 for 2005, MERRA for
2005), allowing us to examine the sensitivity of model $O_3$-CO correlations to input meteorological
data. To understand how various emissions drive global $O_3$-CO correlation patterns, we also
investigated the sensitivity of GMI/MERRA model-calculated $O_3$ and CO concentrations and their
correlations to emission types.

16        We evaluated GMI-simulated tropospheric $O_3$ vertical profiles and tropospheric $O_3$

columns (TOCs) with those from ozonesonde and satellite observations, respectively. To aid in
the evaluation, model simulations of radionuclide tracers ($^{222}$Rn, $^{210}$Pb, and $^7$Be) were used to
illustrate the differences in convection, stratospheric influence, and large-scale subsidence among
three meteorological data sets. Among the three GMI simulations, GMI/GEOS4-simulated $O_3$
concentrations are in best agreement with the observations. GMI/MERRA underestimates $O_3$ in
the NH high-latitude UT due to weak STE, and overestimates $O_3$ in the SH subtropics due to



tropical deep convection being too shallow, which results in less low-$O_3$ air transported from LT

to MT/UT, as well as excessive $NO_x$ emissions from lightning. The latitudinal distribution of

model biases in TOCs relative to satellite observations is consistent with the results from model

evaluations with ozonesonde $O_3$ profiles.

We evaluated GMI simulated $O_3$ and CO concentrations with TES observations at 618 hPa

where TES has most sensitivity. TES observed $O_3$ enhancements over the NH mid-latitudes

(including continental outflow regions), the Middle East, and the subtropical southern Africa and

Atlantic. All simulations well capture the global spatial distribution of $O_3$ at 618 hPa, but appear

to underestimate TES $O_3$ observations over southern Africa and its outflow region. GMI/fvGCM

simulates the highest $O_3$ concentrations at NH mid-/high-latitudes, especially the Asian continent

due to strong STE whereas it simulates the lowest $O_3$ concentrations in the southern tropics and

subtropics due to weak STE and low lightning $NO_X$ emissions. GMI/MERRA simulates the highest

$O_3$ concentrations in the southern subtropics, especially southern Africa due to high lightning $NO_x$

emissions and, to a lesser extent, strong convection. GMI/fvGCM underestimates the $O_3$ minimum

in the tropical western Pacific and eastern Indian Ocean. Considering the positive bias in TES $O_3$

at NH mid-latitudes, GMI/fvGCM appears to overestimate $O_3$ over the East Asian outflow region

due to too fast STE whereas GMI/GEOS4 and GMI/MERRA simulate $O_3$ enhancements

reasonably well in East Asia and its downwind regions. All three simulations significantly

underestimate TES-observed CO enhancements at NH mid-latitudes, but simulate better CO

enhancements over the tropical biomass burning regions.

The three GMI simulations all show strong positive $O_3$-CO correlations at 618 hPa over

the NH mid-latitude continental outflow regions and the SH biomass burning outflow regions, as

shown by TES observations. Generally, positive $O_3$-CO correlations are simulated in downwind



of polluted regions due to photochemical production of $O_3$ from its precursors. However, owing

to significant influences from the stratosphere and subsidence from UT/LS over these regions,

mixing of stratospheric air with polluted (anthropogenic or biomass burning) air masses is

associated with strong positive $O_3$-CO correlations with large $dO_3/dCO$ enhancement ratios.

Strong positive $O_3$-CO correlations are also simulated over the Indonesian biomass burning region

where deep convection occurs, but the $dO_3/dCO$ enhancement ratios are smaller than those in the

NH mid-latitude continental outflow regions. The latter reflects the lower efficiency of $NO_X$

emissions from biomass burning. Strong negative $O_3$-CO correlations over northern and central

Africa, tropical Atlantic, and tropical eastern and western Pacific in all simulations result from

convective transport of biomass burning air masses with low-$O_3$, and consumption of $O_3$ along

with production of CO due to oxidation of biogenic hydrocarbons (e.g., isoprene under low $NO_x$

conditions). The simulated negative $O_3$-CO correlations over the Asian continent, including the

Middle East, are partly attributed to stratospheric influence and/or subsidence from UT/LS. High-

$O_3$ and low-CO associated with stratospherically influenced air lead to strong negative correlations

with large $dO_3/dCO$ ratios. On the other hand, over Southwest China, monsoonal convective lifting

of air masses with high-CO and low-$O_3$ results in negative $O_3$-CO correlations. By contrast, TES

$O_3$ and CO concentrations at 618 hPa either miss such negative correlations (i.e., tropical

convective regions) or only show weak negative correlations over much narrower areas (i.e., the

Tibetan Plateau and northern Africa).

TES-observed $O_3$ and CO concentrations at 618 hPa show highest positive correlations

with large regression slopes over the western Pacific,  and relatively high correlations over North

America, the Middle East, northern South America, central and southern Africa, and continental

outflow regions. Negative correlations are observed in parts of the Asian continent (Tibetan





Plateau), northern Africa, and SH mid-latitudes. All model output sampled along the TES orbit
track capture the observed positive $O_3$-CO correlations over the NH mid-latitude continental
outflow regions, southern Africa, western Indian Ocean, subtropical South Atlantic, northern
South America, and tropical eastern Pacific . While all simulations show strong negative
correlations over the Tibetan Plateau, northern Africa, northern subtropical eastern Pacific, and
Caribbean, TES $O_3$ and CO concentrations at 618 hPa only show weak negative correlations over
much narrower areas (i.e., the Tibetan Plateau and northern Africa).

8       We performed sensitivity simulations with GMI/MERRA to investigate the effect of

individual emission types on model-calculated $O_3$-CO correlations at 618 hPa. Results show that
fossil fuel emissions increase global $O_3$ and CO concentrations and are responsible for the strong
positive correlations over the NH continental outflow regions. Both biomass burning and biogenic
emissions significantly increase global CO concentrations. Biomass burning emissions increase
$O_3$ concentrations in the easterly outflow in the tropical South America and Central Africa, in the
westerly outflow in the southern subtropics, and over Indonesia. Biogenic emissions increase $O_3$
concentrations in the NH subtropics and mid-latitudes, but decrease $O_3$ concentrations in tropical
South America, tropical Africa, and Indonesia. The decreases mainly reflect the fact that $O_3$ is
consumed during the atmospheric oxidation process of isoprene under low $NO_x$ conditions.
Biomass burning emissions are responsible for the positive correlations in the SH mid- and high-
latitudes and negative correlations over part of the tropical western Pacific, Bay of Bengal, NH
subtropical Atlantic, and NH high latitudes. Biogenic emissions have relatively smaller impact on
the correlations than other emissions do, but are largely responsible for the negative $O_3$-CO
correlations over the tropical eastern Pacific. Lightning $NO_x$ emissions lead to large increases in
$O_3$ concentrations at NH subtropics and mid-latitudes, and at SH tropics and subtropics, especially



in the regions of subsidence. We find that lightning $NO_x$ emissions weaken both positive $O_3$-CO
correlations at mid- and high-latitudes and negative correlations in the tropics, and change weak
positive correlations to negative in various areas. This result contrasts with that of previous studies.

4        This study demonstrates the utility of $O_3$-CO correlations to constrain the sources of

tropospheric $O_3$ in global 3-D models. Our model simulations driven by three input meteorological
data sets show significantly different global and regional distributions of $O_3$ and CO concentrations
during boreal summer. For instance, GMI/fvGCM simulations show higher $O_3$ concentrations in
the NH and lower CO concentrations than other simulations. Despite such differences, all
simulations show similar patterns of $O_3$-CO correlations on a global scale. The regional features
of the correlations, however, are often different due to the discrepancies in various meteorological
processes (e.g., convection, STE, subsidence). In particular, GMI/MERRA simulates broader areas
of strong negative $O_3$-CO correlations at 618 hPa in the tropics than GMI/fvGCM and
GMI/GEOS4 do due to stronger tropical convection in the LT/MT. In this sense, $O_3$-CO
correlations can be used to constrain better the sources of regional tropospheric $O_3$ in global models,
especially for convective regions than $O_3$ and CO observations individually. Future work will
examine the driving factors for $O_3$-CO correlations in other seasons.
**Data availability**
A description of the model output and observational data used in this paper can be found in Sect.
2 and they are available upon request by contacting Hongyu Liu (hongyu.liu-1@nasa.gov).





**Acknowledgements.** This work was supported by the NASA Modeling, Analysis, and Prediction
(MAP) program and NASA Atmospheric Composition Modeling and Analysis Program
(ACMAP). NASA Center for Computational Sciences (NCCS) provided supercomputing
resources. TES data products are distributed by NASA Langley Atmospheric Science Data Center.

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



**Table 1**. Global lightning $NO_X$ emissions (Tg N/mon) during May – August in GMI CTM driven
by three meteorological data sets (fvGCM, GEOS4, and MERRA)

|  | May | June | July | August |
|---|---|---|---|---|
| fvGCM | 0.57 (0.03[a]) | 0.65 (0.02) | 0.80 (0.03) | 0.78 (0.05) |
| GEOS4 | 0.64 (0.05) | 0.73 (0.07) | 0.82 (0.03) | 0.72 (0.05) |
| MERRA | 0.49 (0.07) | 0.69 (0.07) | 0.81 (0.03) | 0.78(0.04) |

[a] Values in parenthesis denote lightning $NO_x$ emissions between 10°S and 70°S.






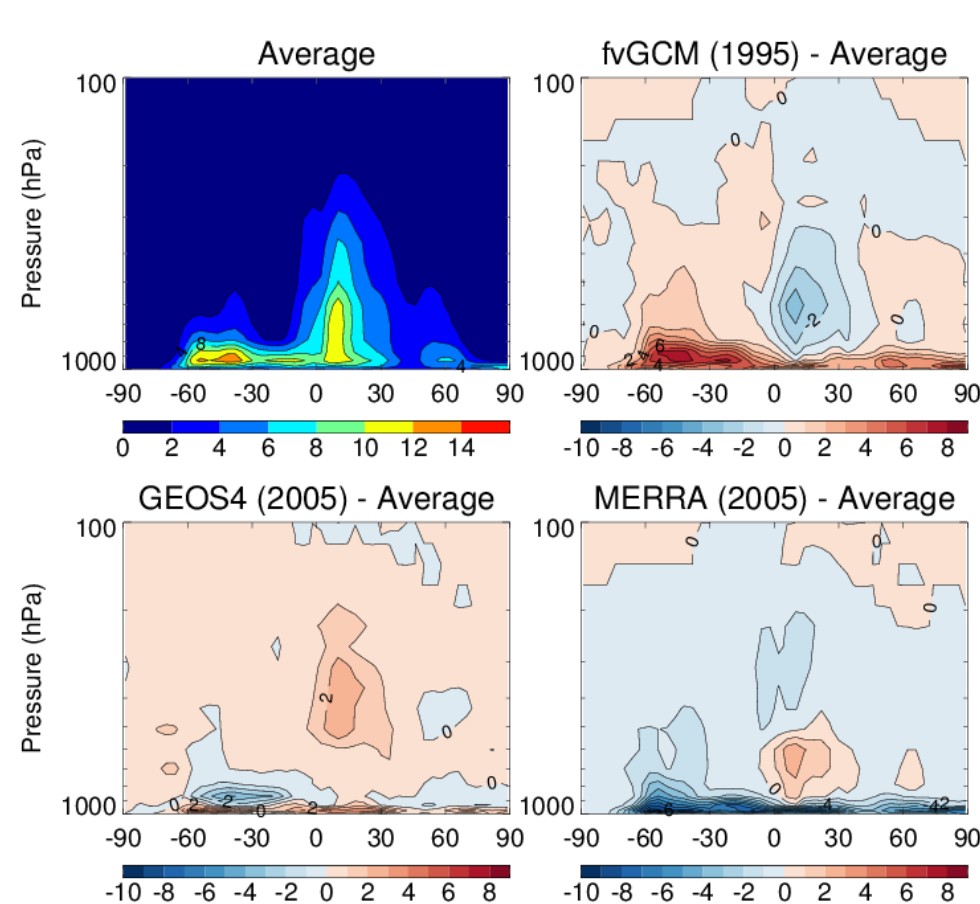

**Figure 1.** Latitude-height cross sections of zonal mean convective mass fluxes during July-August.
The plot shows the values averaged over the fvGCM (1995), GEOS4 (2005), and MERRA (2005)
meteorological data sets, as well as differences from the average.





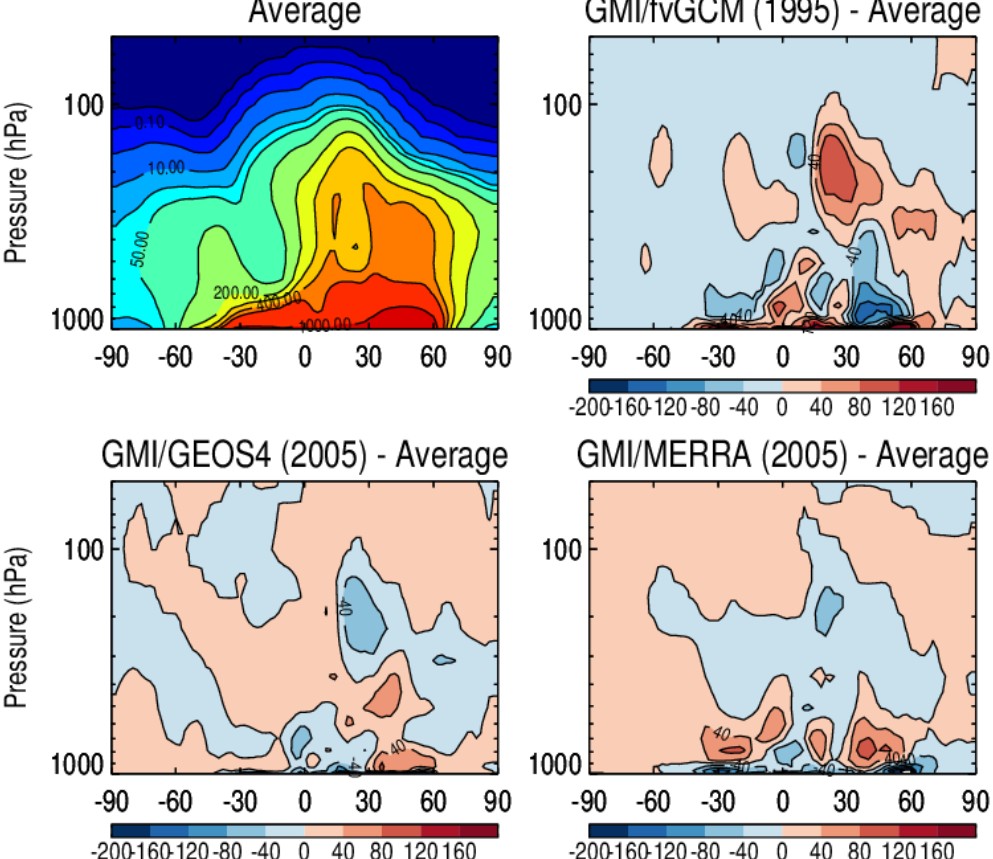

**Figure 2.** Latitude-height cross sections of zonal mean $^{222}$Rn concentrations (mBq SCM$^{-1}$) as
simulated by GMI for July-August. The plot shows the values averaged over three simulations
driven by the fvGCM (1995), GEOS4 (2005), and MERRA (2005) meteorological data sets, as
well as differences of each simulation from the average.





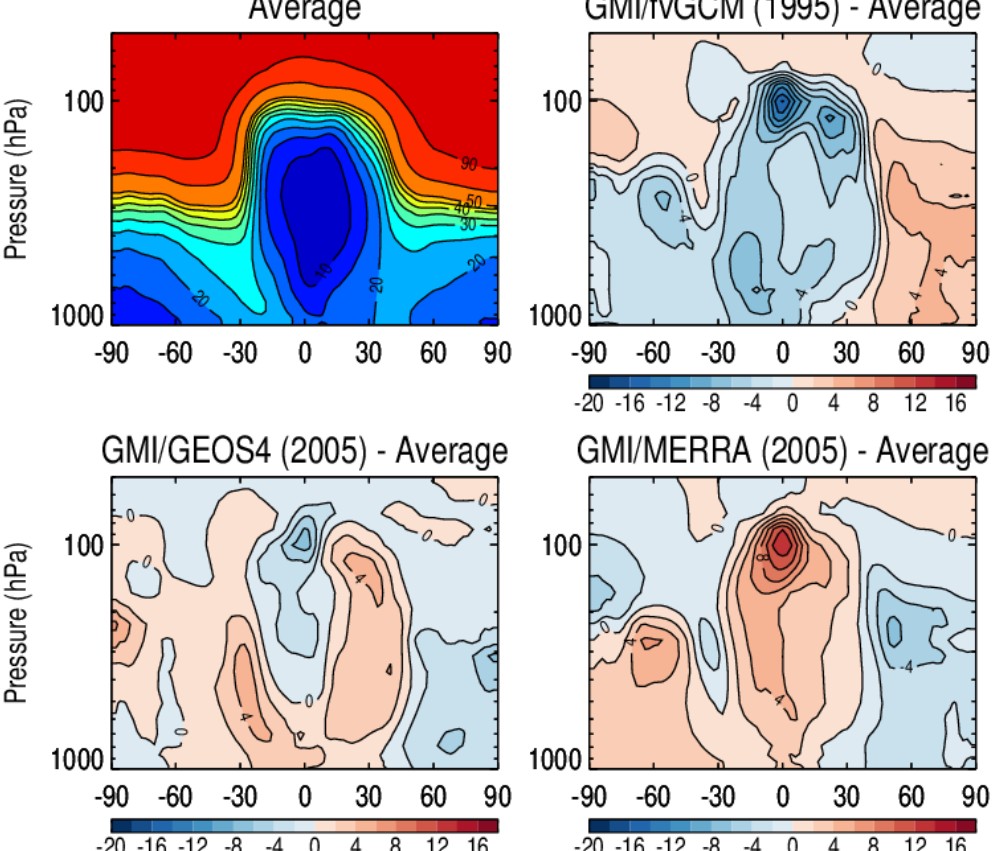

**Figure 3**. Same as **Figure 2**, but for stratospheric fraction (%) of zonal mean tropospheric $^{7}$Be concentrations.



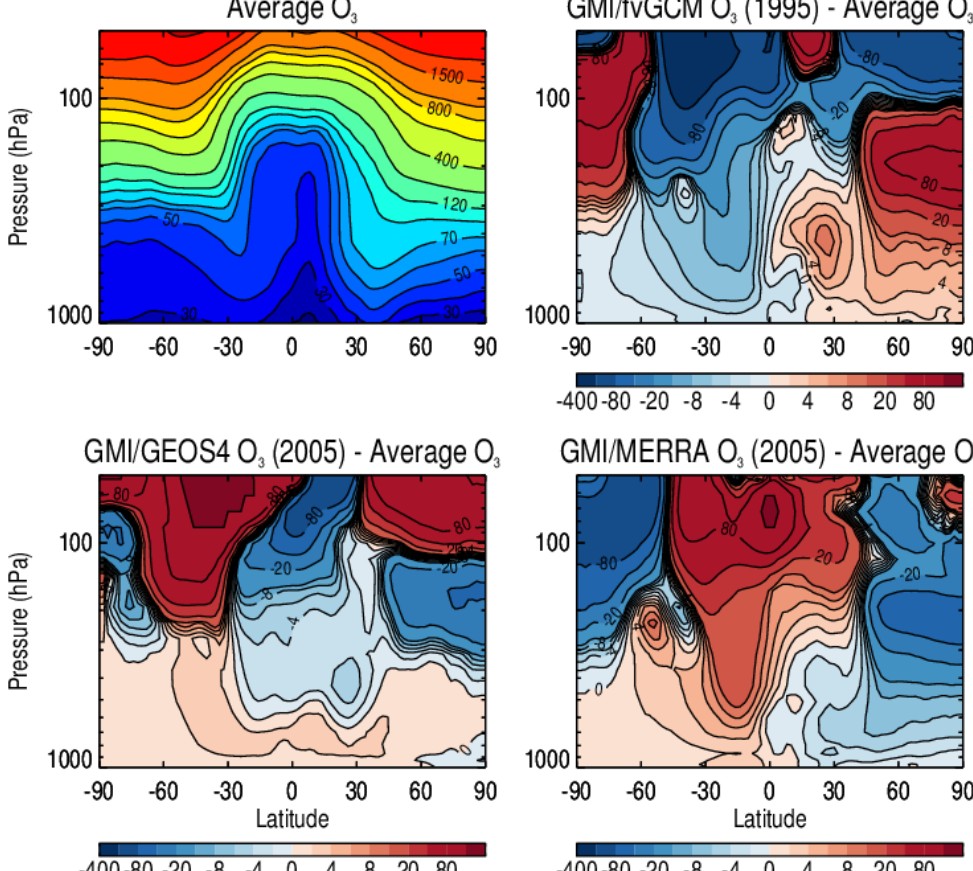

5      **Figure 4**. Same as **Figure 2**, but for zonal mean ozone mixing ratios (ppbv).



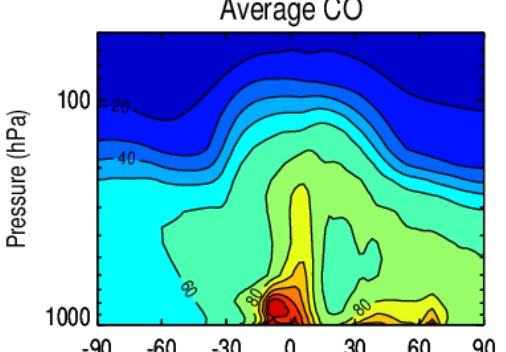
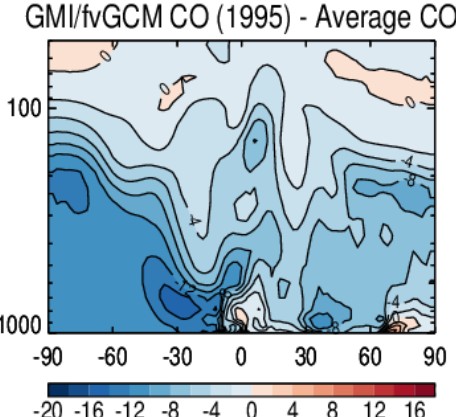

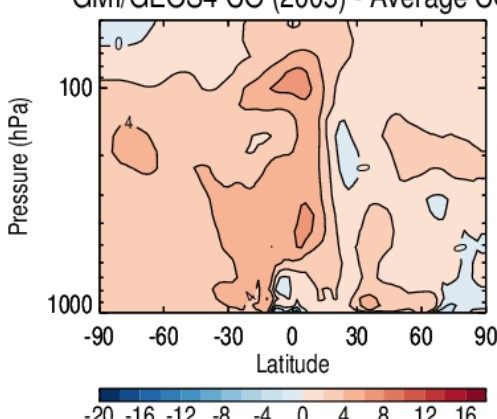
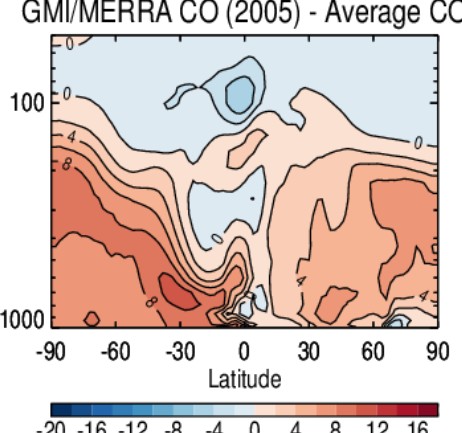

**Figure 5**. Same as **Figure 2,** but for zonal mean CO mixing ratios (ppbv).



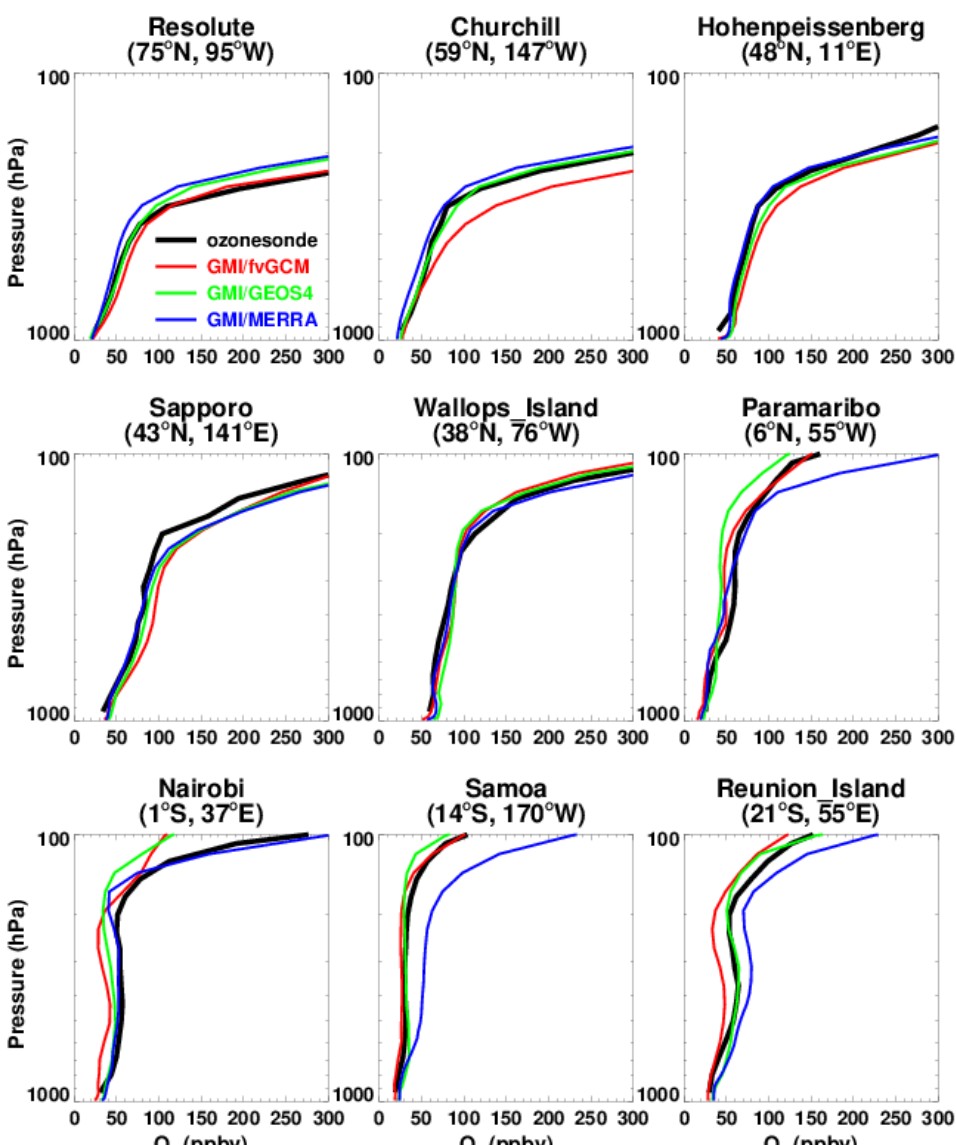

**Figure 6.** Comparisons of GMI simulated tropospheric ozone profiles (color lines) with
ozonesonde observations (black line) for a range of latitudes. The model is driven by the fvGCM,
GEOS4, and MERRA meteorological fields. Values are July–August average.



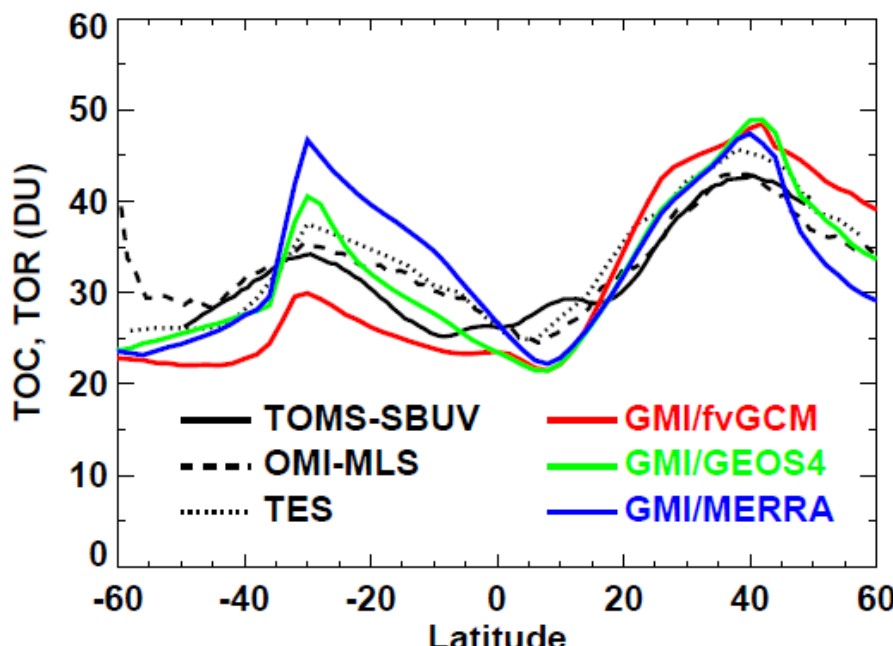

**Figure 7.** GMI simulated zonal mean tropospheric ozone columns (TOCs) compared with tropospheric ozone residuals (TORs) determined from TOMS/SBUV (July–August 1979-2005 multi-year average) (Fishman et al., 2003), OMI/MLS (July–August 2005 average ) (Ziemke et al., 2006), and TOCs determined from TES retrievals (July–August 2005 average).





4  **Figure 8**. July-August mean mixing ratios of $O_3$ and CO (ppbv) at 618 hPa as simulated by GMI
5  CTM driven by three meteorological datasets (1995 for fvGCM, 2005 for GEOS4 and MERRA).



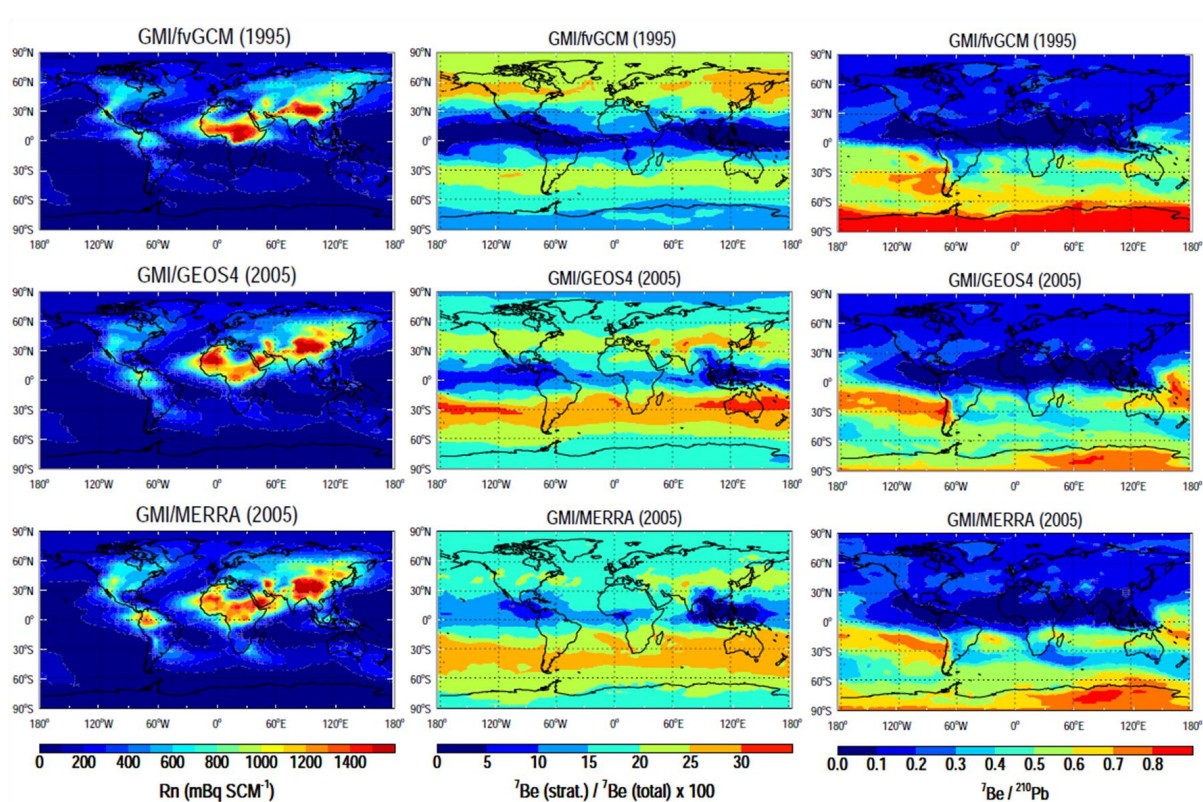

**Figure 9**. Mean $^{222}$Rn concentrations (mBq SCM$^{-1}$) (left column), stratospheric fraction (%) of tropospheric $^{7}$Be concentrations
(middle column), and ratios of $^{7}$Be to $^{210}$Pb volume mixing ratios (right column) at 618 hPa in the GMI model driven by the fvGCM
(1995), GEOS4 (2005), and MERRA (2005) meteorological data sets for the period of July - August.



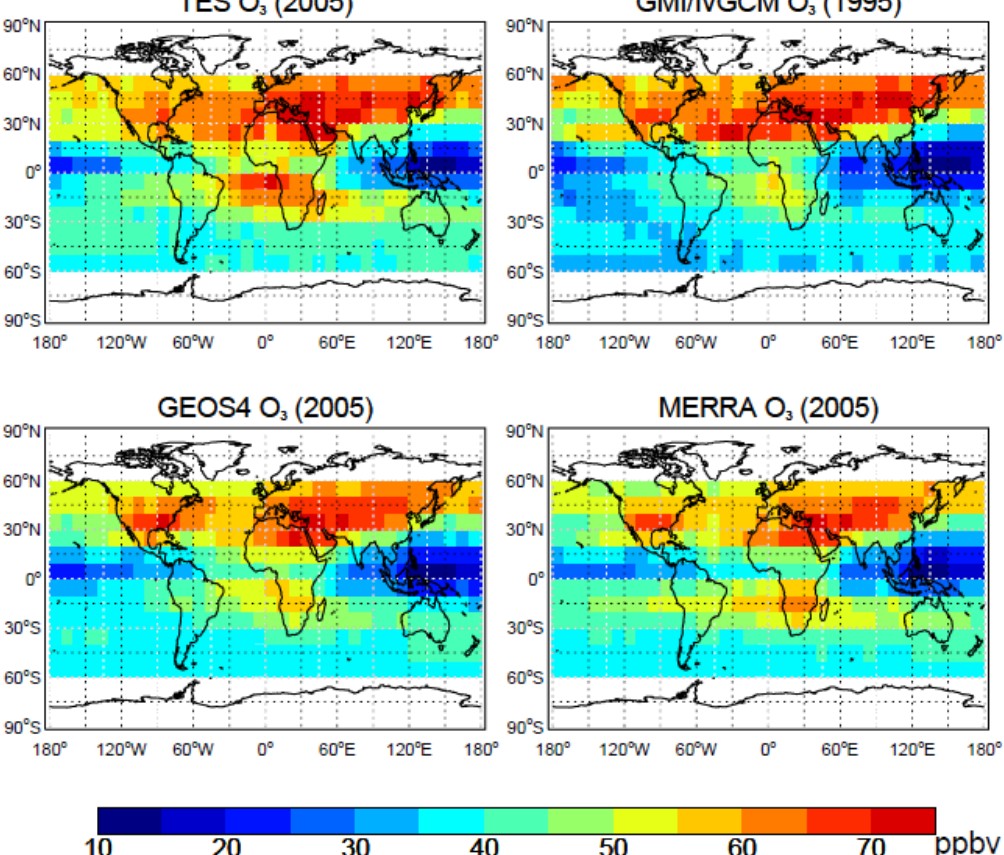

**Figure 10.** Mean mixing ratios of $O_3$ at 618 hPa observed by TES during July- August 2005 and corresponding GMI CTM results with 3-hourly output sampled along the TES orbit tracks. TES averaging kernels and *a priori* were applied to the model output. Results are averaged into $10° \times 10°$ grid cells.



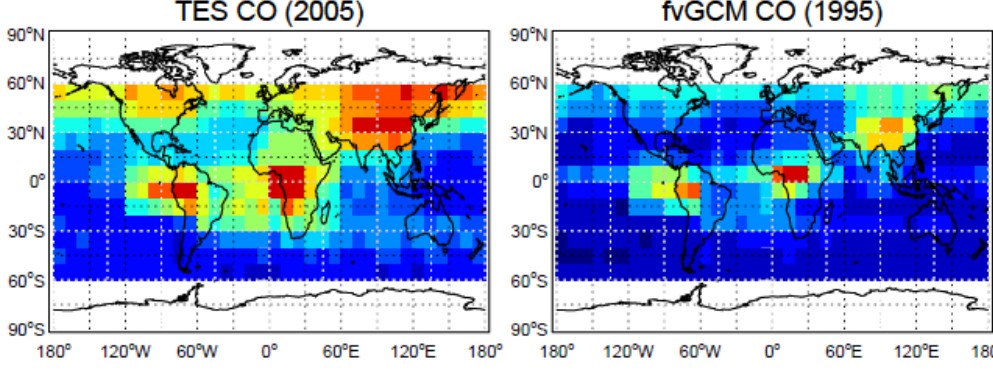

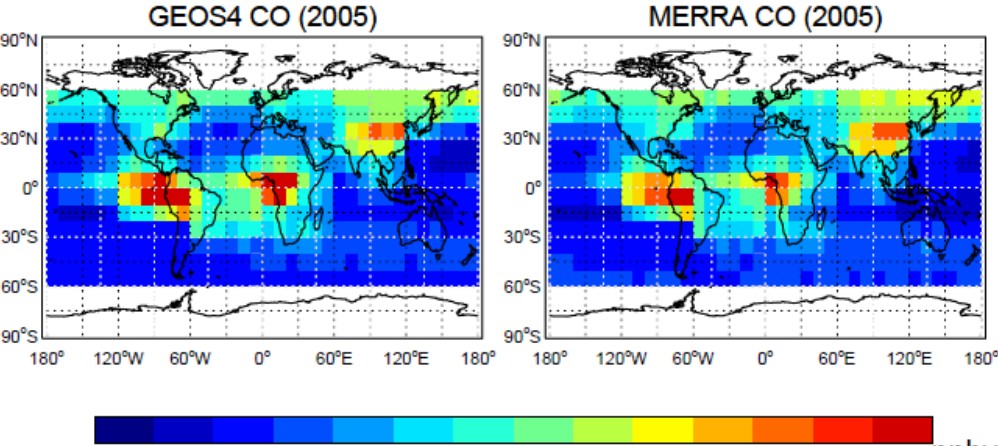

7        **Figure 11.** Same as **Figure 10**, but for CO.





**Figure 12.** $O_3$-CO correlation coefficients (R) and linear regression slopes ($dO_3/dCO$) at 618 hPa in the GMI model driven by the fvGCM (1995), GEOS4 (2005), and MERRA (2005) meteorological fields. Results are calculated in $2° \times 2.5°$ grid cells using 3-hourly model output and the reduced major axis method. White areas denote absolute values of $O_3$-CO correlation coefficients less than 0.2.





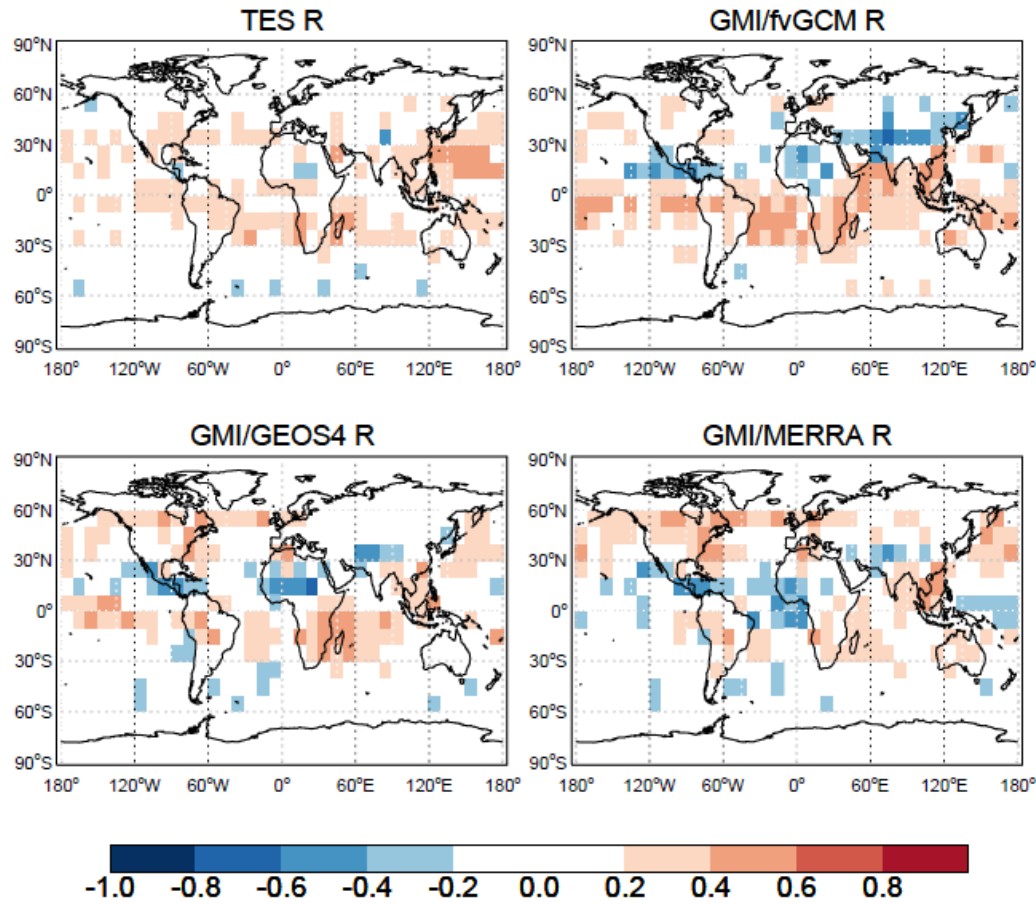

**Figure 13.** $O_3$-CO correlation coefficients (R) at 618 hPa as determined by $O_3$ and CO observations
from TES during July – August 2005, and corresponding GMI CTM results with 3-hourly output
sampled along the TES orbit tracks. TES averaging kernels, spectral errors, and a priori are applied.
Results are calculated in $10° \times 10°$ grid cells. White areas denote absolute values of $O_3$-CO
correlation coefficients less than 0.2.





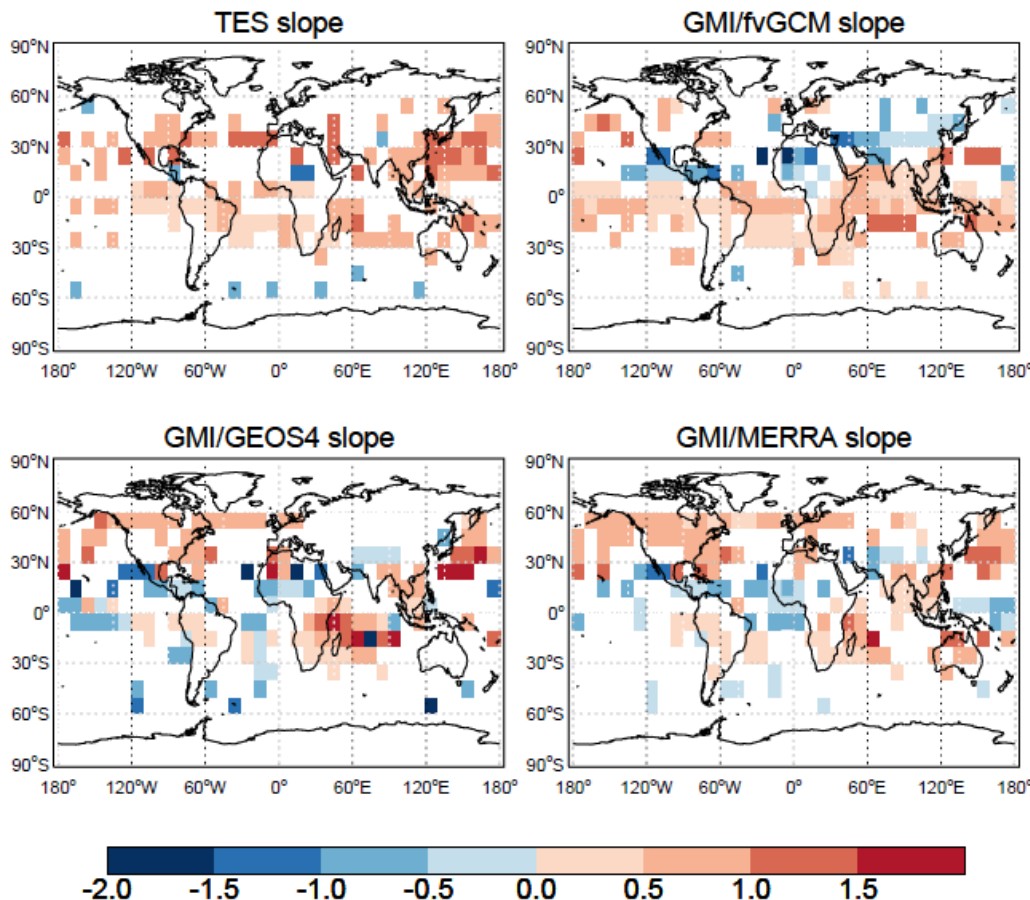

4    **Figure 14.** Same as **Figure 13**, but for linear regression slopes $dO_3/dCO$.





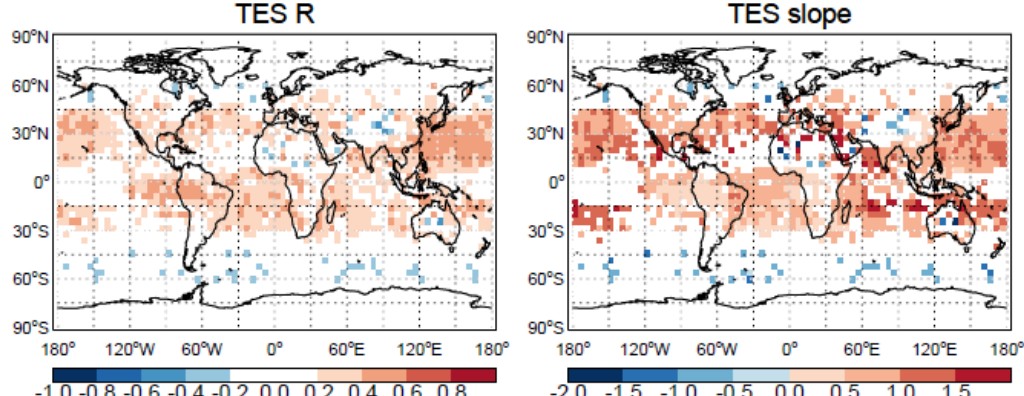

**Figure 15.** $O_3$-CO correlation coefficients (R) and linear regression slopes ($dO_3/dCO$) at 618 hPa
as determined by $O_3$ and CO observations from TES during July–August, 2005 – 2009. Results
are calculated in $4° \times 5°$ grid cells. White areas denote absolute values of $O_3$-CO correlation
coefficients less than 0.2.





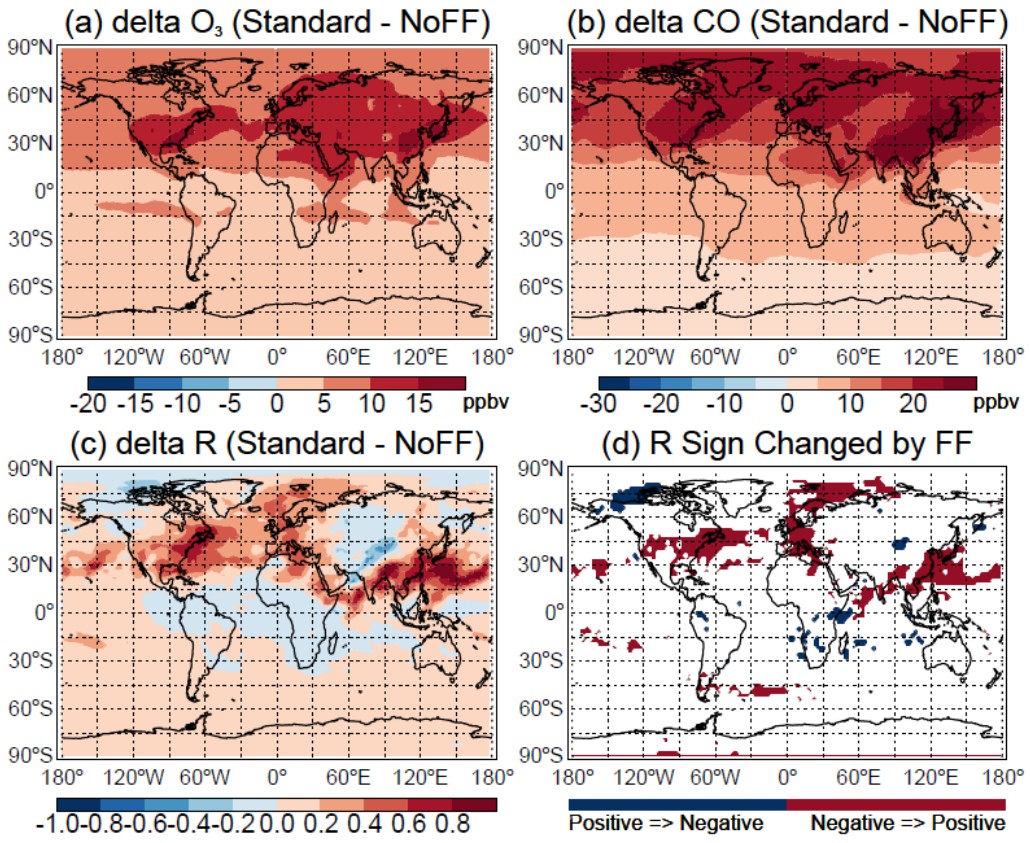

**Figure 16.** Sensitivity of $O_3$, CO, and their correlations to fossil fuel (FF) emissions during July –
August 2005. The plots show the mean differences in (a) $O_3$, (b) CO mixing ratios (ppbv), and (c)
$O_3$-CO correlation coefficients (R) at 618 hPa between the standard GMI/MERRA simulation and
a simulation where fossil fuel emissions are suppressed (NoFF) in the model. Also shown in (d)
are the areas with changed correlation signs. Results are calculated using 3-hourly model output.





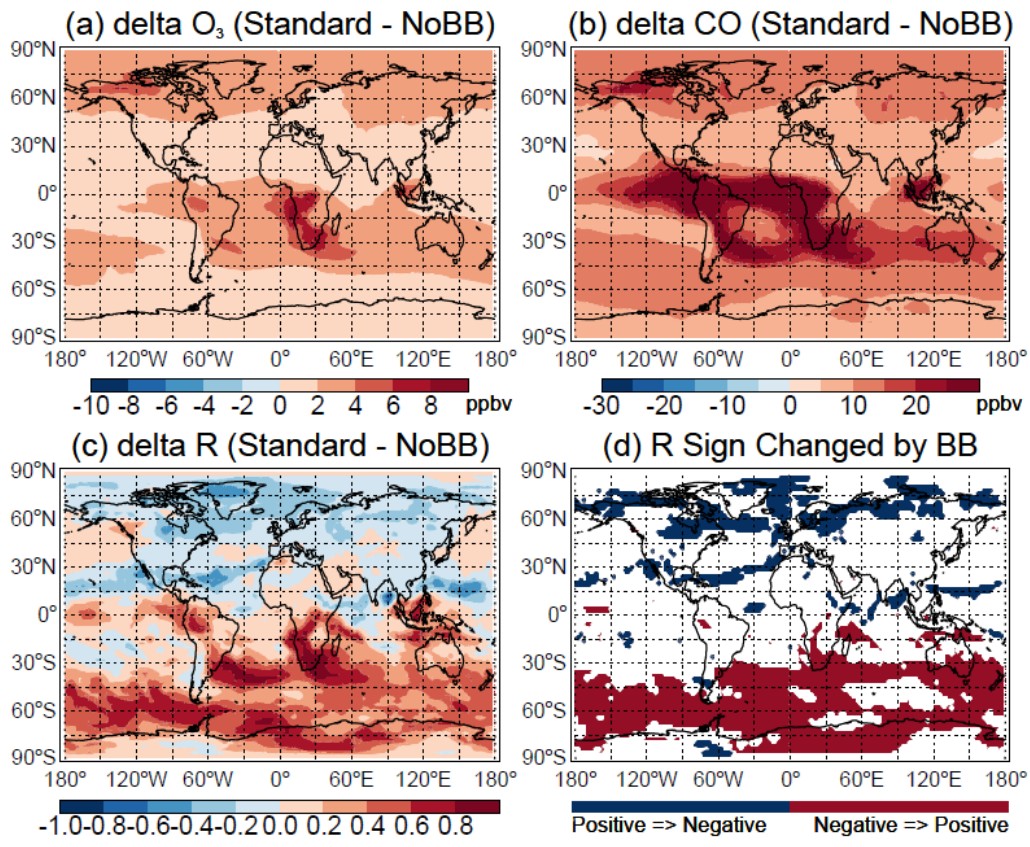

4  **Figure 17.** Same as **Figure 16**, but for the sensitivity of O₃, CO, and their correlations at 618 hPa
5  to biomass burning (BB) emissions.





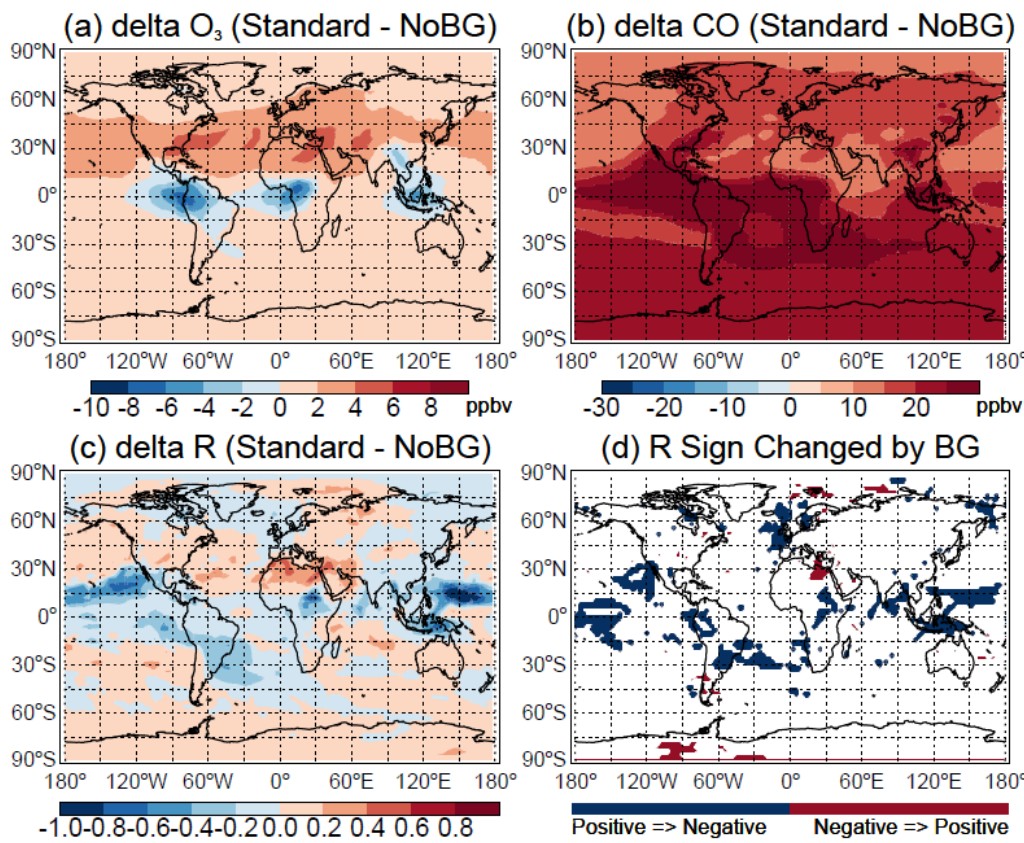

4  **Figure 18.** Same as **Figure 16**, but for the sensitivity of O₃, CO, and their correlations at 618 hPa
5  to biogenic (BG) emissions.





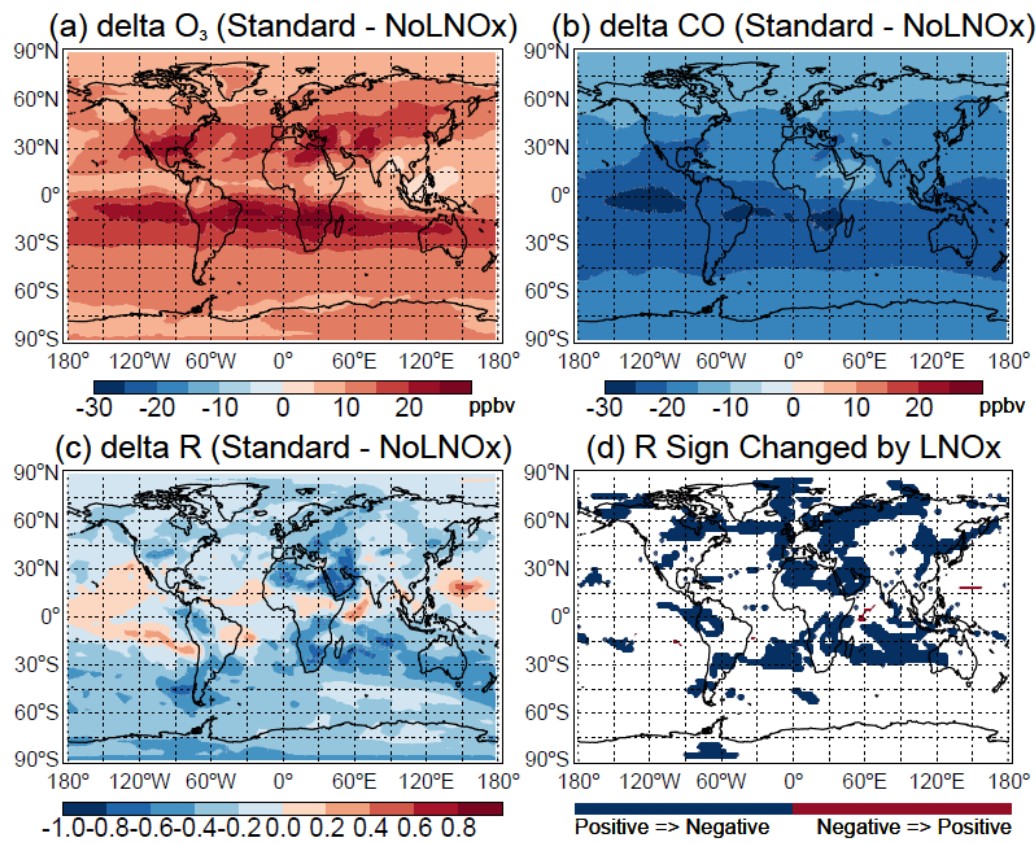

4   **Figure 19.** Same as **Figure 16**, but for the sensitivity of O$_3$, CO, and their correlations at 618 hPa
5   to lightning NO$_X$ (LNO$_X$) emissions.



**Figure 20.** GMI/MERRA-simulated $O_3$-CO correlations (R) at 618 hPa (a) in the standard simulation and (b) - (e) when fossil fuel (FF), biomass burning (BB), biogenic (BG), and lightning NOx ($LNO_X$) emissions are individually suppressed (NoFF, NoBB, NoBG, and $NoLNO_X$, respectively) in the model during July-August 2005. Results are calculated using 3-hourly model output. White areas denote absolute values of $O_3$-CO correlation coefficients less than 0.2.