# Peer review of "Global O3-CO Correlations in a Chemistry and Transport"

_Atmospheric Chemistry and Physics, 2016_

## Referee Comment (RC1) · Dr Huijnen (Referee) · 3 Feb 2017

**General comments**

This is a very well written manuscript describing a thorough analysis of the model $O_3$-CO correlations in the free troposphere, as a means to analyze origins of model biases. The authors analyze various meteorological drivers, and the relative contribution of different emission sources to the modeled correlations, and evaluate them against TES observations at 618hPa.

Overall I have very few comments, hence I recommend this manuscript for publication.

**Specific comments**

In the manuscript I miss a comment regarding the spinup-time of the individual model runs.

Also for GMI/fvGCM you use meteorology from the year 1995, while for the other two simulations you take 2005 meteorology. Did you analyze any potential systematic differences between these two years? I understand this has no impact on the general conclusions drawn from this work.

The authors analyze the impact of three meteorological drivers. Here I think it would be very interesting if they would have included, or will include in future work, ECMWF-based meteorology (ERA-Interim) in their analysis.

The authors blame an over-estimate of $O_3$ in the SH subtropics in GMI/MERRA due to too shallow tropical deep convection. Even though the contribution of this process is clearly illustrated by the [222]Rn-based analysis, I wonder if the conclusion is correct: is there a possibility of compensating errors? Is there independent evidence that MERRA tropical deep convection is too shallow?

The authors also analyze the contribution of STE to tropopspheric $O_3$. Here, it would be useful to see budget numbers of their (annual mean) STE, to be able to intercompare with other systems.

**Technical corrections**

Pp11, line 10: **The** tropopause…

Pp17, line 13: …previously suggested that **the** $O_3$ maximum…

Pp19, line 12: …found **a** multi-model…

Pp29, line 23: …simulated  downwind …

---

## Referee Comment (RC2) · Anonymous Referee #1 · 16 Feb 2017

General

This study explored the ozone-CO correlations on the global scale in boreal summer using a chemical transport model (Global Modeling Initiative (GMI)), driven by three sets of meteorological data: fvGCM with sea surface temperature for 1995, GEOS4-DAS for 2005, and MERRA for 2005. The simulations are compared with the measurements from the Tropospheric Emission Spectrometer (TES) satellite instrument so the model's capability to reproduce the TES data and sensitivity to various meteorological

data were examined. Three radionuclide tracers were simulated as proxies for various transport-related processes to help untangle the simulated ozone-CO correlations and explain the differences. Sensitivity of ozone-CO correlations to various emissions was tested with GMI-MERRA simulations.

This study has addressed an important issue in atmospheric chemistry. The paper is well written with logic flow of text, clear description of the method and assumptions, proper and adequate literature review, and high quality of figures. This study is novel and solid. It offers new insight on global ozone-CO correlations and underlying mechanisms.

Specific

While GMI simulates tropospheric ozone reasonably well, it underestimates tropospheric CO as suggested in this and earlier studies. This underestimation may cause some biases for the ozone-CO correlations presented in this study. Please discuss.

In the model simulations, the anthropogenic emissions are kept the same (using 2005's emissions) for the simulations driven by the three sets of meteorological data. The surface biomass and biogenic emissions are all the same for the three simulations. Therefore, the differences seen in the three simulations are due to different meteorological fields that also cause the differences in lightning emissions. As understandable, the authors placed their focus on the middle troposphere when comparing GMI and TES results because TES data are least biased at these altitudes (Figures 12-15). Therefore, showing NOX emissions from lightning in the middle troposphere horizontally (like Figure 9) would help interpret Figures 12-15.

For the model validation (Figure 6), please provide the values of correlation coefficients, mean biases, and root mean square error so to help evaluate the performance of each simulation quantitatively (in the figure or in a table).

Page 2, Line 13, the authors claimed the simulated ozone-CO correlation patters are

consistent with those derived from TES observations, except in the tropical easterly biomass burning outflow regions. This claim is not be fully supported by Figures 13 and 14. There are large regions with negative correlations in the simulations that are not shown in the TES data. There are other discernible discrepancies between TES and GMI data that should be mentioned and discussed.

Page 21, Line 3-4: The authors stated: "Strong positive O3-CO correlations are present in all simulations at 618 hPa over Indonesia (Figure 12)". Over the entire Indonesia? The positive correlations appear only over western Indonesia where simulations show high CO.

Remove an extra comma near the ends for Luo et al. (2007 a and b) and Mao, H., and Talbot, R. (2004) in References.

Word "Figure" may not be in bold in the final version.

---

## Author Comment (AC2) · 16 May 2017

We thank Referee #1 for the review and very useful suggestions. Our responses are
itemized below.

"General. This study explored the ozone-CO correlations on the global scale in boreal
summer using a chemical transport model (Global Modeling Initiative (GMI)), driven
by three sets of meteorological data: fvGCM with sea surface temperature for 1995,

[Figure]

GEOS4-DAS for 2005, and MERRA for 2005. The simulations are compared with the measurements from the Tropospheric Emission Spectrometer (TES) satellite instrument so the model's capability to reproduce the TES data and sensitivity to various meteorological data were examined. Three radionuclide tracers were simulated as proxies for various transport-related processes to help untangle the simulated ozone-CO correlations and explain the differences. Sensitivity of ozone-CO correlations to various emissions was tested with GMI-MERRA simulations. This study has addressed an important issue in atmospheric chemistry. The paper is well written with logic flow of text, clear description of the method and assumptions, proper and adequate literature review, and high quality of figures. This study is novel and solid. It offers new insight on global ozone-CO correlations and underlying mechanisms."

Reply – Thanks for the comments.

"Specific. While GMI simulates tropospheric ozone reasonably well, it underestimates tropospheric CO as suggested in this and earlier studies. This underestimation may cause some biases for the ozone-CO correlations presented in this study. Please discuss."

Reply – Good point. We have revised the first paragraph of Section 5 (P19): "In this section, we examine O3 and CO relationships at 618hPa in GMI CTM. We interpret GMI simulated O3-CO correlations and their slopes in the context of emissions, photochemical transformation, and transport (e.g., convection, STE, and large-scale subsidence), using model meteorological data and radionuclide simulations. We then evaluate them with those derived from TES satellite observations. Note that the model underestimate of CO concentrations does not significantly affect the calculated O3-CO correlations although it may cause biases in the regression slopes due to the association of the latter with ozone production efficiency".

"In the model simulations, the anthropogenic emissions are kept the same (using 2005's emissions) for the simulations driven by the three sets of meteorological data.

The surface biomass and biogenic emissions are all the same for the three simulations. Therefore, the differences seen in the three simulations are due to different meteorological fields that also cause the differences in lightning emissions. As understandable, the authors placed their focus on the middle troposphere when comparing GMI and TES results because TES data are least biased at these altitudes (Figures 12-15). Therefore, showing NOX emissions from lightning in the middle troposphere horizontally (like Figure 9) would help interpret Figures 12-15."

Reply – As discussed on P15 (L15-20, Table 1), all simulations show similar lightning NOx (LNOx) emissions during July-August. However, GMI/fvGCM shows a factor of $\sim$ 2.5 lower LNOx emissions than GMI/GEOS4 and GMI/MERRA during May-June. The July-August LNOx emissions do not explain the discrepancy in the simulated ozone, and therefore we decide not to add a plot of LNOx emissions. We have added a sentence in the text (P15) as below: "It is noted that all simulations show hot spots of LNOx emissions over the central and eastern US, central Africa, and west Tibetan plateau (not shown)."

"For the model validation (Figure 6), please provide the values of correlation coefficients, mean biases, and root mean square error so to help evaluate the performance of each simulation quantitatively (in the figure or in a table)."

Reply – Thanks for the suggestion. We have added a new table (Table 2) and a sentence in the text (P15): "The mean differences between simulated O3 and ozonesonde observations at 500 hPa (MT) and 200 hPa (UT), respectively, are listed in Table 2."

"Page 2, Line 13, the authors claimed the simulated ozone-CO correlation patters are consistent with those derived from TES observations, except in the tropical easterly biomass burning outflow regions. This claim is not be fully supported by Figures 13 and 14. There are large regions with negative correlations in the simulations that are not shown in the TES data. There are other discernible discrepancies between TES and GMI data that should be mentioned and discussed."

[Figure]

Reply – Thanks for pointing this out. We have revised the text to "Despite the fact that the three simulations show significantly different global and regional distributions of O3 and CO concentrations, they show similar patterns of O3-CO correlations on a global scale. All model simulations sampled along the TES orbit track capture the observed positive O3-CO correlations in the Northern Hemisphere mid-latitude continental outflow and the Southern Hemisphere subtropics. While all simulations show strong negative correlations over the Tibetan Plateau, northern Africa, northern subtropical eastern Pacific, and Caribbean, TES O3 and CO concentrations at 618 hPa only show weak negative correlations over much narrower areas (i.e., the Tibetan Plateau and northern Africa). Discrepancies in regional O3-CO correlation patterns in the three simulations may be attributed to differences in convective transport, stratospheric influence, and subsidence, among other processes."

"Page 21, Line 3-4: The authors stated: "Strong positive O3-CO correlations are present in all simulations at 618 hPa over Indonesia (Figure 12)". Over the entire Indonesia? The positive correlations appear only over western Indonesia where simulations show high CO."

Reply – Indeed. We have revised the sentence to "Strong positive O3-CO correlations are present in all simulations at 618 hPa over western and central Indonesia (Figure 12), reflecting convective transport of biomass burning CO (Figure 8) and photochemical production of O3 from its precursors."

"Remove an extra comma near the ends for Luo et al. (2007 a and b) and Mao, H., and Talbot, R. (2004) in References."

Reply – Done.

"Word "Figure" may not be in bold in the final version."

Reply – Corrected.

---

## Author Response (AR1)

**Reply to Referee #1's comments:**

We thank Referee #1 for the review and very useful suggestions. Our responses are itemized
below.

*"General. This study explored the ozone-CO correlations on the global scale in boreal summer*
*using a chemical transport model (Global Modeling Initiative (GMI)), driven by three sets of*
*meteorological data: fvGCM with sea surface temperature for 1995, GEOS4-DAS for 2005, and*
*MERRA for 2005. The simulations are compared with the measurements from the Tropospheric*
*Emission Spectrometer (TES) satellite instrument so the model's capability to reproduce the TES*
*data and sensitivity to various meteorological data were examined. Three radionuclide tracers*
*were simulated as proxies for various transport-related processes to help untangle the simulated*
*ozone-CO correlations and explain the differences. Sensitivity of ozone-CO correlations to*
*various emissions was tested with GMI-MERRA simulations. This study has addressed an*
*important issue in atmospheric chemistry. The paper is well written with logic flow of text, clear*
*description of the method and assumptions, proper and adequate literature review, and high*
*quality of figures. This study is novel and solid. It offers new insight on global ozone-CO*
*correlations and underlying mechanisms."*

**Reply** – Thanks for the comments.

*"Specific. While GMI simulates tropospheric ozone reasonably well, it underestimates*
*tropospheric CO as suggested in this and earlier studies. This underestimation may cause some*
*biases for the ozone-CO correlations presented in this study. Please discuss."*

**Reply** – Good point. We have revised the first paragraph of Section 5 (P19): "In this section, we
examine $O_3$ and CO relationships at 618hPa in GMI CTM. We interpret GMI simulated $O_3$-CO
correlations and their slopes in the context of emissions, photochemical transformation, and
transport (e.g., convection, STE, and large-scale subsidence), using model meteorological data
and radionuclide simulations. We then evaluate them with those derived from TES satellite
observations. Note that the model underestimate of CO concentrations does not significantly
affect the calculated $O_3$-CO correlations although it may cause biases in the regression slopes
due to the association of the latter with ozone production efficiency".

*"In the model simulations, the anthropogenic emissions are kept the same (using 2005's*
*emissions) for the simulations driven by the three sets of meteorological data. The surface*
*biomass and biogenic emissions are all the same for the three simulations. Therefore, the*
*differences seen in the three simulations are due to different meteorological fields that also cause*
*the differences in lightning emissions. As understandable, the authors placed their focus on the*
*middle troposphere when comparing GMI and TES results because TES data are least biased at*
*these altitudes (Figures 12-15). Therefore, showing NOX emissions from lightning in the middle*
*troposphere horizontally (like Figure 9) would help interpret Figures 12-15.*"

**Reply** – As discussed on P15 (L15-20, Table 1), all simulations show similar lightning $NO_x$
($LNO_x$) emissions during July-August. However, GMI/fvGCM shows a factor of ~ 2.5 lower
LNOx emissions than GMI/GEOS4 and GMI/MERRA during May-June. The July-August $LNO_x$
emissions do not explain the discrepancy in the simulated ozone, and therefore we decide not to
add a plot of $LNO_x$ emissions. We have added a sentence in the text (P15) as below: "It is noted
that all simulations show hot spots of $LNO_x$ emissions over the central and eastern US, central
Africa, and west Tibetan plateau (not shown)."

*"For the model validation (Figure 6), please provide the values of correlation coefficients, mean*
*biases, and root mean square error so to help evaluate the performance of each simulation*
*quantitatively (in the figure or in a table)."*

**Reply** – Thanks for the suggestion. We have added a new table (Table 2) and a sentence in the
text (P15): "The mean differences between simulated $O_3$ and ozonesonde observations at 500
hPa (MT) and 200 hPa (UT), respectively, are listed in Table 2."

*"Page 2, Line 13, the authors claimed the simulated ozone-CO correlation patters are consistent*
*with those derived from TES observations, except in the tropical easterly biomass burning*
*outflow regions. This claim is not be fully supported by Figures 13 and 14. There are large*
*regions with negative correlations in the simulations that are not shown in the TES data. There*
*are other discernible discrepancies between TES and GMI data that should be mentioned and*
*discussed."*

**Reply** – Thanks for pointing this out. We have revised the text to "Despite the fact that the three simulations show significantly different global and regional distributions of $O_3$ and CO concentrations, they show similar patterns of $O_3$-CO correlations on a global scale. All model simulations sampled along the TES orbit track capture the observed positive $O_3$-CO correlations in the Northern Hemisphere mid-latitude continental outflow and the Southern Hemisphere subtropics. While all simulations show strong negative correlations over the Tibetan Plateau, northern Africa, northern subtropical eastern Pacific, and Caribbean, TES $O_3$ and CO concentrations at 618 hPa only show weak negative correlations over much narrower areas (i.e., the Tibetan Plateau and northern Africa). Discrepancies in regional $O_3$-CO correlation patterns in the three simulations may be attributed to differences in convective transport, stratospheric influence, and subsidence, among other processes."

*"Page 21, Line 3-4: The authors stated: "Strong positive O3-CO correlations are present in all simulations at 618 hPa over Indonesia (Figure 12)". Over the entire Indonesia? The positive correlations appear only over western Indonesia where simulations show high CO."*

**Reply** – Indeed. We have revised the sentence to "Strong positive $O_3$-CO correlations are present in all simulations at 618 hPa over western and central Indonesia (Figure 12), reflecting convective transport of biomass burning CO (Figure 8) and photochemical production of $O_3$ from its precursors."

*"Remove an extra comma near the ends for Luo et al. (2007 a and b) and Mao, H., and Talbot, R. (2004) in References."*

**Reply** – Done.

*"Word "Figure" may not be in bold in the final version."*

**Reply** – Corrected.

**Reply to Dr. Vincent Huijnen's comments:**

We thank Dr. Vincent Huijnen for very helpful comments. Our responses are itemized below.

*"General comments. This is a very well written manuscript describing a thorough analysis of the model O3-CO correlations in the free troposphere, as a means to analyze origins of model biases. The authors analyze various meteorological drivers, and the relative contribution of different emission sources to the modeled correlations, and evaluate them against TES observations at 618hPa. Overall I have very few comments, hence I recommend this manuscript for publication."*

*"Specific comments. In the manuscript I miss a comment regarding the spinup-time of the individual model runs.*

**Reply** – Thanks for pointing this out. Now we state in the text: "All standard and perturbation full-chemistry simulations for July-August as presented in this paper were conducted with a 6-month spinup." (Section 2.1.1), and "All simulations of radionuclide tracers were conducted with a 5-year spinup in order for $^{210}$Pb to reach an equilibrium in the stratosphere." (Section 2.1.3).

*"Also for GMI/fvGCM you use meteorology from the year 1995, while for the other two simulations you take 2005 meteorology. Did you analyze any potential systematic differences between these two years? I understand this has no impact on the general conclusions drawn from this work."*

**Reply** – We did not examine any potential systematic differences between 1995 and 2005 meteorology. GMI/fvGCM was driven with the output from the fvGCM general circulation model (with sea surface temperature for 1995), which was intended to represent only the contemporary climatological state of the atmosphere. Indeed, this does not affect the general conclusion of this study. We have modified the text (P 9) to: "We drive the GMI CTM with three meteorological datasets from: the free-running NASA Global Modeling and Assimilation Office (GMAO) finite-volume General Circulation Model (*fvGCM with sea surface temperature for*

*1995*), the Goddard Earth Observing System Data Assimilation System Version 4 (GEOS4-
DAS) for 2005, and the Modern-Era Retrospective Analysis for Research and Applications
(MERRA) for 2005."

*"The authors analyze the impact of three meteorological drivers. Here I think it would be very*
*interesting if they would have included, or will include in future work, ECMWF-based*
*meteorology (ERA-Interim) in their analysis."*

**Reply** – Indeed, it would be very interesting to include the ERA-Interim meteorology in future
work.  Now we state in the text (P32-33): "Future work, where additional meteorological
archives (e.g., GFDL AM3, ECMWF ERA-Interim) may also be incorporated, should examine
the driving factors for $O_3$-CO correlations in other seasons."

*"The authors blame an over-estimate of O3 in the SH subtropics in GMI/MERRA due to too*
*shallow tropical deep convection. Even though the contribution of this process is clearly*
*illustrated by the 222Rn-based analysis, I wonder if the conclusion is correct: is there a*
*possibility of compensating errors? Is there independent evidence that MERRA tropical deep*
*convection is too shallow?"*

**Reply** – Point well taken. This overestimate also involves factors other than too shallow tropical
deep convection, for which we did not find independent evidence. We have revised the statement
in the Conclusion section to: "Among the three GMI simulations, GMI/GEOS4-simulated $O_3$
concentrations are in best agreement with the observations. GMI/MERRA underestimates $O_3$ in
the NH high-latitude UT due to weak STE, and overestimates $O_3$ in the SH subtropics. The latter
is due to a combination of excessive influences from lightning $NO_x$ emissions and STE (or
subsidence), as well as the shallower convection resulting in less low-$O_3$ air lifted from the LT to
MT/UT."

*"The authors also analyze the contribution of STE to tropospheric O3. Here, it would be useful*
*to see budget numbers of their (annual mean) STE, to be able to intercompare with other*
*systems."*

**Reply** – Unfortunately, our model analysis focused on July-August and the annual mean STE
fluxes of ozone are not available.

*"Technical corrections.  Pp11, l i ne 10: The tropopause...; Pp17, line 13: ...previously*
*suggested that the O3 maximum...; Pp19, line 12: ...found a multi-mode l...; Pp29, l i ne 23:*
*...simulated  downwind ..."*

**Reply** – Corrected.

**Revised text with track changes**

**(next page)**

[revised manuscript text omitted]